# An NMDAR positive and negative allosteric modulator series share a binding site and are interconverted by methyl groups

Riley Perszyk[1], Brooke M Katzman[2], Hirofumi Kusumoto[3], Steven A Kell[2], Matthew P Epplin[2], Yesim A Tahirovic[2], Rhonda L Moore[2], David Menaldino[2], Pieter Burger[2], Dennis C Liotta[2], Stephen F Traynelis[1]*

[1]Department of Pharmacology, Emory University, Atlanta, United States; [2]Chemistry, Emory University, Atlanta, United States; [3]Pharmacology, Emory University, Atlanta, United States

**Abstract** N-methyl-D-aspartate receptors (NMDARs) are an important receptor in the brain and have been implicated in multiple neurological disorders. Many non-selective NMDAR-targeting drugs are poorly tolerated, leading to efforts to target NMDAR subtypes to improve the therapeutic index. We describe here a series of negative allosteric NMDAR modulators with submaximal inhibition at saturating concentrations. Modest changes to the chemical structure interconvert negative and positive modulation. All modulators share the ability to enhance agonist potency and are use-dependent, requiring the binding of both agonists before modulators act with high potency. Data suggest that these modulators, including both enantiomers, bind to the same site on the receptor and share structural determinants of action. Due to the modulator properties, submaximal negative modulators in this series may spare NMDAR at the synapse, while augmenting the response of NMDAR in extrasynaptic spaces. These modulators could serve as useful tools to probe the role of extrasynaptic NMDARs.

DOI: https://doi.org/10.7554/eLife.34711.001

*For correspondence:
strayne@emory.edu

## Introduction

N-methyl-D-aspartate receptors (NMDARs) are a subtype of ionotropic glutamate receptors that are broadly expressed in the brain and are important for normal development, neuronal plasticity and memory formation (*Traynelis et al., 2010*; *Paoletti et al., 2013*). NMDARs contribute a slow, $Ca^{2+}$ permeable component to fast excitatory neurotransmission that is voltage dependent by virtue of its sensitivity to pore block by extracellular $Mg^{2+}$ (*Traynelis et al., 2010*). The typical NMDAR is comprised of 2 GluN1 and 2 GluN2 subunits, creating the potential for diversity given that there are eight splice variants of GluN1 and 4 independent genes encoding GluN2 subunits (A-D) (*Traynelis et al., 2010*). The regulation and functional properties of the NMDAR are controlled by the subunits incorporated into the receptor (*Monyer et al., 1994*; *Vicini et al., 1998*; *Vance et al., 2012*). NMDARs are expressed at most excitatory synapses, and are also found in peri- and extrasynaptic locations (*Sans et al., 2000*; *Steigerwald et al., 2000*; *Groc et al., 2006*; *Traynelis et al., 2010*). Although NMDARs have been studied extensively, there are still important questions about the different roles that NMDARs may play given differences in subunit composition and synaptic localization (*Hardingham and Bading, 2010*; *Gladding and Raymond, 2011*; *Paoletti et al., 2013*). Pharmacological approaches can be useful for probing these questions, but tool compounds to differentiate these NMDAR subtypes have been limited (*Ogden and Traynelis, 2011*;

**eLife digest** The neurons in the brain form networks that can change in response to experience, causing new connections to form between certain neurons and breaking the connections between others. This remodeling process underlies learning and memory. However, in certain neurological disorders, such as schizophrenia and epilepsy, these networks are disrupted and no longer work correctly.

A receptor protein called the NMDA receptor plays an important role in reshaping the networks of neurons. Chemicals called neurotransmitters that are released by one neuron bind to and activate NMDA receptors on a neighbouring neuron to communicate with it. This activation encourages new connections to form between neurons.

Drugs that alter the activity of the NMDA receptor could potentially act as treatments for neurological conditions that disrupt how the networks of neurons work. However, few have been approved for use in patients because most of the potential drug compounds investigated so far produce severe side effects.

Perszyk et al. have now identified a new group of compounds that can potentially alter the activity of NMDA receptors without fully blocking the response, potentially eliminating the unwanted side effects. The compounds were tested on frog egg cells and human embryonic kidney cells that had been engineered to produce NMDA receptors. Small changes to the chemical structure of these compounds could switch their effect from increasing to decreasing the activity of the receptor. The compounds only interact with active receptors that have a neurotransmitter bound to them, and compete with each other to bind to the receptors.

Perszyk et al. also found that some compounds behaved differently depending on how active the NMDA receptor was. These compounds could potentially be used to sense the activity in a network of neurons, which opens up new options for treating neurological conditions that affect the networks.

Further experiments are now required to see how these compounds affect the activity of NMDA receptors in neurons and living animals. The range of effects produced by the compounds studied by Perszyk et al. suggests that other related compounds may have different effects on receptor activity. Future work could investigate the properties of these compounds to see if they could treat a different set of neurological disorders.

DOI: https://doi.org/10.7554/eLife.34711.002

Monaghan et al., 2012; Santangelo et al., 2012; Strong et al., 2014; Zhu and Paoletti, 2015). Whereas NMDAR receptors have been implicated in many neurological diseases, there remains a dearth of clinically-approved drugs that target NMDARs (Kalia et al., 2008; Traynelis et al., 2010; Paoletti et al., 2013; Strong et al., 2014).

Multiple endogenous and exogenous modulatory sites in the NMDAR recently have been described. In addition, nearly complete NMDAR structures obtained using crystallographic approaches are now available (Karakas and Furukawa, 2014; Lee et al., 2014), and these data, together with recent cryo-EM structures in the receptor super family (Twomey and Sobolevsky, 2018; Twomey et al., 2017a; Twomey et al., 2017b), illustrate the overall topography of the NMDAR and suggest a mechanism for some allosteric modulators, such as GluN2B-selective ifenprodil (Tajima et al., 2016; Zhu et al., 2016). However, a complete understanding of how other modulatory sites operate and can be exploited has been elusive. Advances in the understanding of specific roles of particular NMDAR subunits have led to renewed interest in targeted therapeutic intervention, and recent work has yielded a growing tool box of novel ligands that act on NMDARs with diverse sites and mechanisms (Ogden and Traynelis, 2011; Monaghan et al., 2012; Hackos and Hanson, 2017). To date, there are positive and negative modulators that bind to the amino-terminal domain (ATD), agonist binding domain (ABD), or transmembrane domain (TMD). Each new class of compounds discovered enriches our understanding of the function of NMDAR and how these receptors can be modulated. The information gained from these modulators has the potential to provide insight into both NMDAR function and therapeutic strategies to treat complex neurological diseases.

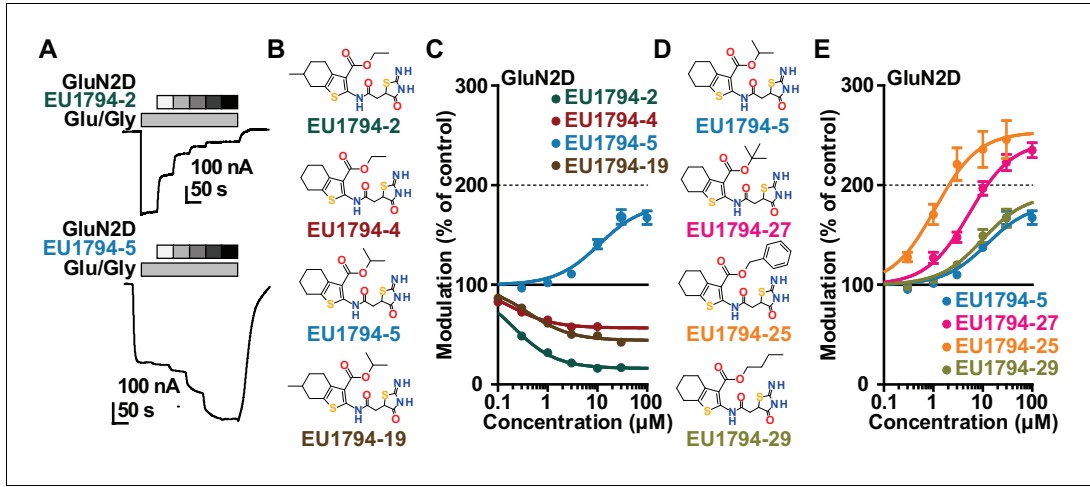

**Figure 1.** The EU1794 series of NMDAR NAMs can be converted to PAMs with subtle structural modifications. (**A**) Representative concentration-response experiments for a negative (**EU1794-2**, above) and positive (**EU1794-5**, below) allosteric modulator using two-electrode voltage-clamp recordings of GluN1/GluN2D NMDARs activated by saturating concentrations of agonist. 100/30 µM glutamate/glycine application is represented by the grey bar and increasing concentrations of modulator are shown by grey scale boxes (0.3, 1, 3, 10, 30 µM for **EU1794-2** and 1, 3, 10, 30, 100 µM for **EU1794-5**). (**B**) Chemical structures of the NAMs **EU1794-2**, **EU1794-4**, **EU1794-19** and PAM **EU1794-5** are given, which show all permutations of two methyl substitutions. This series illustrates how subtle changes in chemical structure interconvert modulator activity. (**C**) The concentration-response curves for the compounds shown in (**B**) acting on GluN1/GluN2D NMDARs activated by maximally effective concentrations of co-agonists and fitted by the Hill equation. (**D**) Chemical structures of **EU1794** PAMs. (**E**) Concentration-response curves for **EU1794** PAMs on the responses of GluN1/GluN2D NMDARs to maximally effective concentrations of co-agonists. See *Table 1* for fitted EC$_{50}$ values at all diheteromeric NMDARs. Data represent 4–18 oocytes recorded in at least two independent experiments.

DOI: https://doi.org/10.7554/eLife.34711.003

This study describes a series of compounds that includes both positive allosteric modulators (PAMs) as well as negative allosteric modulators (NAMs) of NMDAR function (*Katzman et al., 2015*). A set of aliphatic substitutions off an ester linkage to the scaffold interconverts these compounds between positive and negative allosteric modulators. Remarkably, the difference between the positive and negative modulation to the same chemical scaffold was influenced by the addition or removal of individual methyl groups. These modulators appear to bind to a shared site to bring about opposing actions, and share mechanistic features, such as agonist dependence and enhancement of agonist potency. The spectrum of properties in this series of modulators could serve as useful tool compounds for probing the role of NMDARs in circuits in both healthy brain and in neuropathological situations.

## Results

### Identification of a new class of positive allosteric modulators of NMDAR function

We have previously described the structure-activity relationship (SAR) of a series of negative allosteric pan-NMDAR modulators that contained an amidothiophene core, the most potent possessing a tetrahydrobenzothiophene, with different alkyl and aryl substitutions connected via an ester linkage at the 3-position (*Katzman et al., 2015*). These compounds inhibited the response to saturating concentration of co-agonists at NMDARs expressed in *Xenopus laevis* oocyte experiments, typified by **EU1794-2** (compound 4 in *Katzman et al., 2015*). We describe here a new set of closely related analogues that can either potentiate or inhibit responses depending on subtle changes in structure and agonist concentration. As shown in *Figure 1* and *Table 1*, **EU1794-4** contained an ethyl ester similar to **EU1794-2** and lacked a methyl substituent on the tetrahydrobenzothiophene core.

**Table 1.** The effect of alkyl ester and tetrahydrobenzothiophene ring substitutions on the potency and efficacy of **EU1794** analogues.

| Compound | $R_1$ | R1 volume ($\text{Å}^3$) | $R_2$ | EC$_{50}$ ($\mu$M) [conf. int.]* Maximal Degree of Modulation (% of control)[†] | | | |
|---|---|---|---|---|---|---|---|
| | | | | GluN2A | GluN2B | GluN2C | GluN2D |
| EU1794-4 | Et | 45.15 | H | 2.2 [1.8, 2.8] 32 ± 3% | 2.6 [1.4, 4.8] 67 ± 4% | 0.42 [0.28, 0.61] 52 ± 2% | 0.36 [0.29, 0.45] 51 ± 3% |
| EU1794-2 | Et | 45.15 | Me | 0.60 [0.44, 0.82] 6 ± 2% | 1.2 [0.8, 1.9] 10 ± 3% | 0.26 [0.21, 0.31] 14 ± 1% | 0.20 [0.17, 0.25] 14 ± 1% |
| EU1794-5 | iPr | 62.16 | H | 18 [9.0, 36] 36 ± 5% | 2.5 [1.4, 4.4] 71 ± 4% | 4.0 [3.5, 4.6] 140 ± 3% | 9.3 [8.1, 11] 169 ± 7% |
| EU1794-19 | iPr | 62.16 | Me | 1.1 [0.5, 2.3] 19 ± 6% | 1.1 [0.9, 1.3] 43 ± 2% | 0.74 [0.57, 0.96] 32 ± 3% | 0.43 [0.32, 0.59] 46 ± 2% |
| EU1794-27 | tBu | 79.23 | H | 7.4 [5.3, 10] 52 ± 7% | 1.4 [1.2, 1.7] 130 ± 3% | 2.8 [2.5, 3.2] 230 ± 10% | 2.4 [1.8, 3.3] 250 ± 8% |
| EU1794-25 | Bn | 98.87 | H | - | 1.0 [0.9, 1.2] 220 ± 13% | 0.77 [0.56, 1.0] 180 ± 12% | 1.0 [0.95, 1.1] 250 ± 19% |
| EU1794-29 | n-Bu | 79.12 | H | 3.7 [1.7, 8.0] 59 ± 5% | 1.4 [1.1, 1.8] 140 ± 3% | 3.8 [2.6, 5.7] 130 ± 7% | 5.7 [5.1, 6.4] 166 ± 8% |

*EC$_{50}$ values for potentiation of responses to saturating glutamate and glycine (100 $\mu$M, 30 $\mu$M) were obtained by least-squares fitting of data from individual experiments by the Hill equation. EC$_{50}$ values are given as the mean with the 95% confidence interval determined from log(EC$_{50}$).

[†]The maximal degree of modulation is given as mean ±SEM. For all compounds, data are from 4 to 18 oocytes recorded in at least two independent experiments. Data were not fit (shown as -) if the response recorded at 30 $\mu$M of test compound did not differ by more than 15% from control.

DOI: https://doi.org/10.7554/eLife.34711.004

**EU1794-4** potently inhibited GluN1/GluN2D with a substantial residual current remaining at saturating concentrations (*Figure 1B,C*, *Table 1*). Interestingly, ester substitutions that had a larger calculated functional group volume than the ethyl ester in **EU1794-4** potentiated NMDAR responses to saturating concentrations of glutamate and glycine. For example, the isopropyl ester (**EU1794-5**) potentiated GluN1/GluN2D responses to nearly 200% of control (*Figure 1A–C*). Restoration of the methyl to the tetrahydrobenzothiophene core restored negative allosteric modulation (**EU1794-19**, *Figure 1B,C*). Thus, the direction of modulation (positive or negative) could be determined by the size of the alkyl ester and substitution to the tetrahydrobenzothiophene core.

The size of the ester-linked substituent controlled the extent and potency of positive modulation. The *t*-butyl ester **EU1794-27** potentiated GluN1/GluN2D NMDARs with greater efficacy and higher potency, increasing responses to maximally effective glutamate and glycine to 250% of control with an EC$_{50}$ value of 2.4 $\mu$M (*Figure 1D,E*). The benzyl ester (**EU1794-25**) also strongly potentiated responses with a potency similar to **EU1794-27** (*Figure 1D,E*). However, **EU1794-29**, which had a longer substitution, *n*-butyl, reduced both the maximal potentiation and potency, suggesting an optimal substituent size and shape (*Figure 1D,E*).

## Allosteric modulation of agonist potency by NAMs and PAMs

The tetrahydrobenzothiophene-containing NAMs reported here inhibit all diheteromeric NMDARs without substantial subunit selectivity, similar to those that we previously described (*Katzman et al., 2015*). By contrast, the novel PAMs described here show distinct GluN2 dependence (*Table 1*). All PAMs are active at GluN1/GluN2C and GluN1/GluN2D, but do not enhance the response of GluN1/GluN2A to maximally effective concentrations of agonist, in some cases resulted in inhibition of these responses. Most PAMs were also capable of potentiating GluN1/GluN2B (*Table 1*, *Figure 2A*). As seen with other series of NMDAR PAMs (*Malayev et al., 2002*; *Horak et al., 2004*; *Horak et al., 2006*; *Hackos et al., 2016*; *Wang et al., 2017*), allosteric modulation can be dependent on agonist concentrations. Thus, we assessed the ability of **EU1794-27** to modulate NMDAR responses activated by sub-saturating co-agonist concentrations, clearly exemplified by modulation

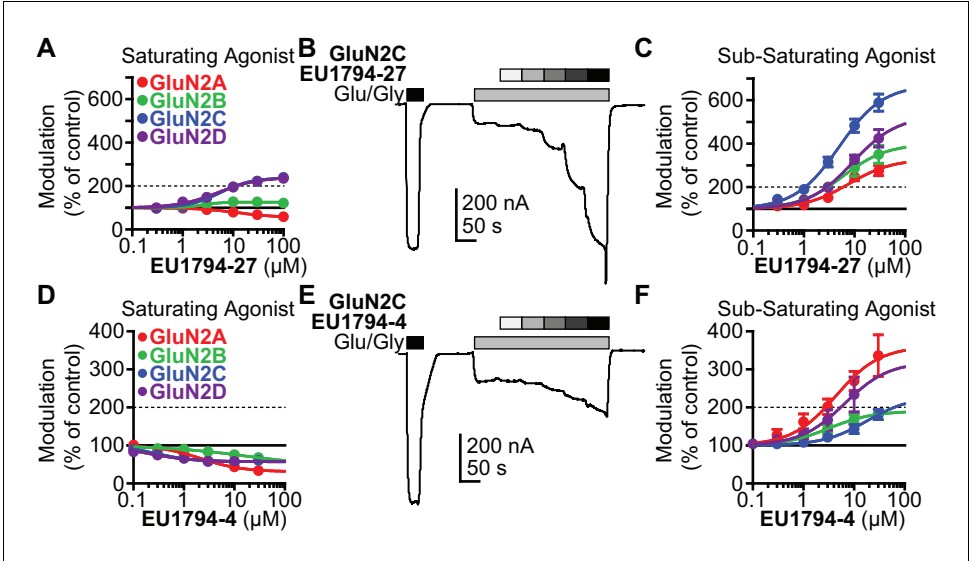

**Figure 2.** Modulation of both **EU1794** PAMs and NAMs is dependent on agonist concentration. (A) The concentration-response relationship for **EU1794-27** at diheteromeric NMDARs activated by 100/30 µM glutamate/glycine (data for GluN1/GluN2D shown in *Figure 1E* is included here for clarity). GluN1/GluN2C and GluN1/GluN2D (which are superimposed) are potentiated to a greater extent than GluN1/GluN2B; there appears to be slight inhibition of GluN1/GluN2A. (B) A representative concentration-response recording of **EU1794-27** at GluN1/GluN2B activated by sub-saturating concentrations of agonist. The initial response was activated using saturating 100/30 µM glutamate/glycine (black box) followed by 1/0.3 µM glutamate/glycine (grey box, roughly resulting in an activity state that was 20% of the maximal response) administered in the absence and presence of increasing concentrations of **EU1794-27** (0.3, 1, 3, 10, 30 µM). (C) Concentration-response relationship for **EU1794-27** at diheteromeric NMDARs activated by sub-saturating concentration of agonist (glutamate/glycine concentrations used for GluN1/GluN2A were 2/0.6 µM, GluN1/GluN2B and GluN1/GluN2C were 1/0.3 µM, and GluN1/GluN2D were 0.6/0.2 µM). Note potentiation of all subunits. (D–F) Similar experiments as in A-C, but using the NAM **EU1794-4** at saturating (D) or sub-saturating (E,F) concentrations of agonist (glutamate/glycine concentrations used for GluN1/GluN2A, GluN1/GluN2B and GluN1/GluN2C were 1/0.3 µM, and GluN1/GluN2D were 0.3/0.09 µM); data for GluN1/GluN2D presented in *Figure 1C* are included here for comparison. Note the inhibition by the NAM at saturating agonist concentration (D) compared to potentiation in submaximal concentrations of agonist in (E,F). Data represent 4–13 oocytes recorded in two independent experiments.

DOI: https://doi.org/10.7554/eLife.34711.005

The following figure supplement is available for figure 2:

**Figure supplement 1.** EU1794-2 actions possess modest agonist concentration-dependence.

DOI: https://doi.org/10.7554/eLife.34711.006

---

of GluN1/GluN2C (*Figure 2B*, *Table 2*). In these conditions, **EU1794-27** positively modulated all NMDAR subtypes and enhanced previously potentiated subtypes to a greater extent than in saturating agonist (compare *Figure 2A and C*). We hypothesized that this agonist-dependence was due to **EU1794-27** altering the agonist potency. Therefore, we determined the glutamate and glycine $EC_{50}$ values in the absence and presence of **EU1794-27** (*Figure 3A,B*, *Table 3*, *Figure 3—figure supplement 1*). **EU1794-27** produced modest but significant decreases (higher potency) in the $EC_{50}$ values for both glutamate and glycine at all NMDARs (*Table 3*, *Figure 3—figure supplement 1*). Given this effect by the PAMs in this series, we subsequently considered whether the NAMs in the series shared this mechanism. We selected **EU1794-4** for use in evaluating actions on agonist potency since it retains a large steady-state current even for receptors that have bound **EU1794-4** (saturated inhibition is between 40–70% of control, *Figure 2D*). We co-applied **EU1794-4** during responses stimulated by sub-saturating concentrations of agonist (glutamate and glycine concentrations that resulted approximately in a $EC_{20}$ response), which resulted in positive modulation (*Figure 2E,F*, *Table 2*). Furthermore, the negative modulator **EU1794-4** enhanced the glutamate and glycine potencies (*Figure 3C,D*, *Table 4*, *Figure 3—figure supplement 1*). Interestingly, positive modulation elicited by 10 µM **EU1794-4** is dependent on the sub-saturating agonist concentrations used

**Table 2.** Effect of EU1794-4 and EU1794-27 at sub-saturated NMDAR responses.

| | EC$_{50}$ ($\mu$M) [conf. int.]* Maximal Degree of Modulation (% of control)[†] | | | |
| --- | --- | --- | --- | --- |
| | GluN2A | GluN2B | GluN2C | GluN2D |
| EU1794-4 | 3.9 [0.78, 20] 500 ± 32% | 4.3 [1.3, 14] 210 ± 6.8% | 15 [6.5, 35] 220 ± 20% | 14 [6.5, 29] 540 ± 190% |
| EU1794-27 | 8.1 [5.7, 12] 340 ± 32% | 6.3 [5.0, 8.0] 400 ± 50% | 5.0 [3.7, 6.6] 680 ± 50% | 8.3 [5.3, 13] 580 ± 86% |

*EC$_{50}$ values of modulator action on responses to sub-saturating glutamate/glycine concentrations were obtained by least-squares fitting of data from individual experiments by the Hill equation. For EU1794-4 modulation of GluN1/GluN2A, GluN1/GluN2B and GluN1/GluN2C, glutamate/glycine concentrations were 1/0.3 $\mu$M; for GluN1/GluN2D glutamate/glycine were 0.3/0.09 $\mu$M. For EU1794-27 potentiation, glutamate/glycine concentrations were 2/0.6 $\mu$M (GluN1/GluN2A), 1/0.3 $\mu$M (GluN1/GluN2B, GluN1/GluN2C), and 0.6/0.2 $\mu$M (GluN1/GluN2D). EC$_{50}$ for potentiation values are given as the mean with the 95% confidence interval determined from log(EC$_{50}$).

[†]The extent of modulation is given as a percent of the control response in the absence of test compound. The maximal degree of modulation is given as mean ±SEM. For all compounds, data are from 4 to 13 oocytes recorded in two independent experiments.

DOI: https://doi.org/10.7554/eLife.34711.009

(**Table 5**). EU1794-4 inhibited GluN1/GluN2C responses to saturating concentrations of agonist to 54% of control, and positively modulated equivalent EC$_{30}$ responses (equal effective concentrations of glutamate and glycine concentrations that when applied resulted in a 30% of a maximal response) to 140% of control, and positively modulated average equivalent EC$_{3}$ responses by 660% (**Table 5**). We did not observe augmentation of sub-saturating agonist responses by EU1794-2, most likely due to its greater extent of inhibition (**Figure 2—figure supplement 1**). However, the degree of inhibition produced by EU1794-2 was reduced on sub-saturating responses at GluN1/GluN2C from 16% to 45% of control (**Table 5**).

## PAMs and NAMs display both glutamate and glycine dependence

We studied NMDARs expressed in HEK293 cells to investigate the time course of modulator action. Concentration-dependent association of EU1794-2, EU1794-4 and EU1794-27 and concentration-independent disassociation were evaluated by co-applying modulator with glutamate and glycine (**Figure 4A,B**). These modulators were tested on GluN1/GluN2A and GluN1/GluN2D to see whether there were differences in modulation given the distinct properties of these NMDAR subtypes. We analyzed the concentration-dependence of the exponential time course describing the onset of action during co-application with saturating concentrations of glutamate plus glycine to determine modulator association and dissociation rates. From these we calculated the kinetically-determined affinity constant (K$_d$), which we found to be similar to the EC$_{50}$ values determined from concentration-response experiments. EU1794-2 K$_d$ was 1.1 $\mu$M at GluN1/GluN2D (**Figure 4A,B**). Complex actions of EU1794-4 and EU1794-27 were observed, with two temporally-distinct phases of modulation evident for the association of these modulators (**Figure 4A**). Both compounds produced a rapid inhibition followed by a slowly developing potentiating phase for EU1794-27. The rapid association rate determined during the rapid inhibition produced by EU1794-4 was approximately three times faster than EU1794-2; similarly, the dissociation rate was also faster for EU1794-4 than EU1794-2. Quantitative analysis of the slower phases was challenging due to its lower signal-to-noise ratio. Likewise, the rapid inhibitory phase was also difficult to measure for EU1794-27 because its rapid time course was convolved with the potentiation time course. Additionally, the depotentiation time course is preceded by a transient enhancement of the current response, followed by a relaxation to the pre-modulation level. The time course for potentiation and depotentiation were best fit with an exponential function summed with an additional linear component after the rapid phase subsided. K$_d$ determined from the association and dissociation rates of EU1794-4 was 4.2 $\mu$M and EU1794-27 was 4.3 $\mu$M (**Figure 4B**). All modulatory effects were independent of voltage (**Figure 4—figure supplement 1A**). The steady state modulator responses from HEK293 cells approximately match the concentration-response relationship determined from TEVC recordings from *X. laevis* oocytes (**Figure 4A**, *inset graphs*). The time-course of modulator binding of EU1794-2, EU1794-4 and

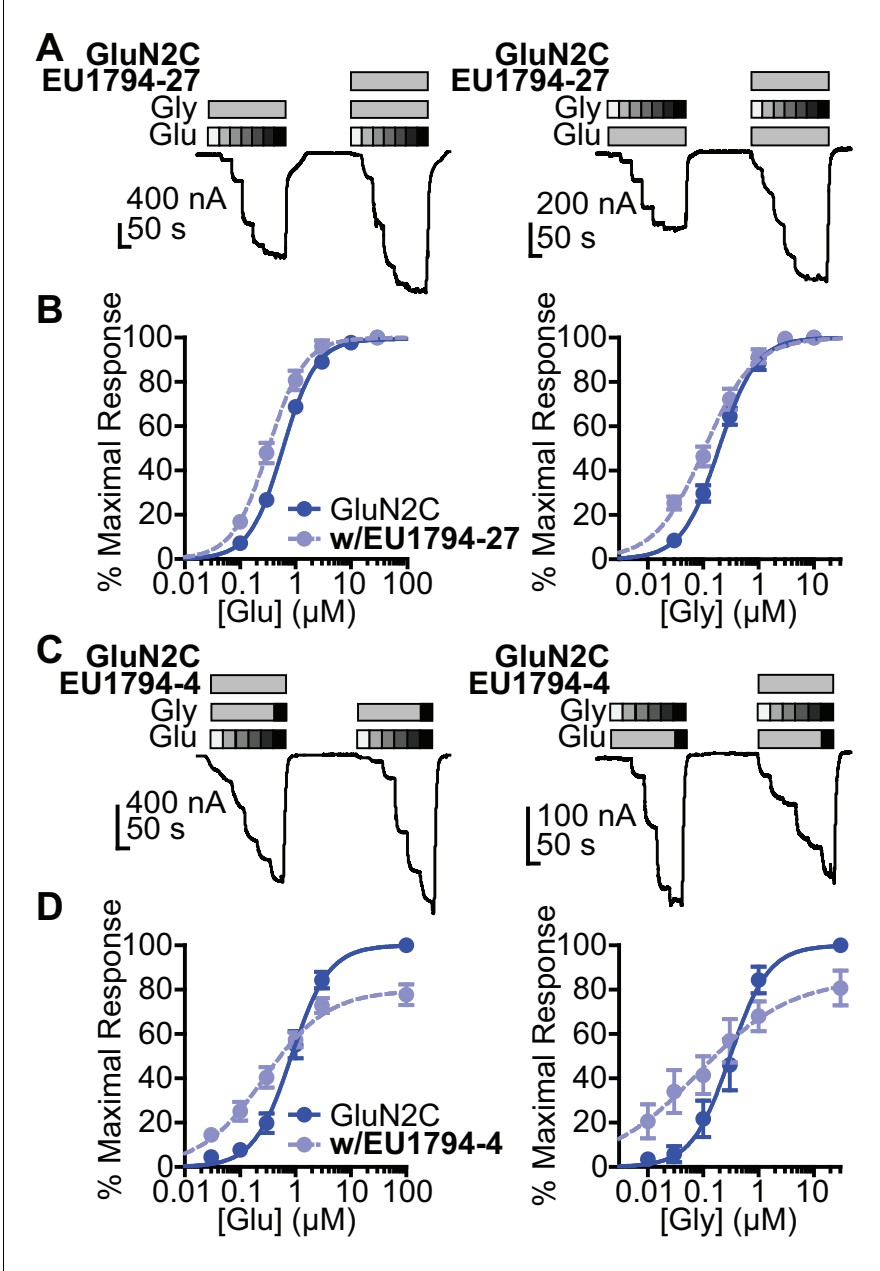

**Figure 3.** EU1794 PAMs and NAMs enhance glutamate and glycine potency. (A) Representative glutamate (left) and glycine (right) concentration-response curves of oocytes expressing GluN1/GluN2C with and without **EU1794-27** (10 µM, grey box). The glutamate concentration-response curve (0.1, 0.3, 1, 3, 10, 30 µM) was generated with 30 µM glycine and the glycine concentration-response curve (0.03, 0.1, 0.3, 1, 3, 10 µM) was recorded in 100 µM glutamate. (B) Mean concentration-response data for glutamate (left) and glycine (right) at GluN1/GluN2C (see *Figure 2—figure supplement 1* for other subunits). Each curve was normalized to the maximal response in the absence of **EU1794-27** to illustrate the actions of **EU1794-27** on agonist $EC_{50}$. (C,D) A similar set of experiments as in (A,B), but with **EU1794-4** are shown (10 µM, grey box). Note the variation of the order of **EU1794-4** application in experimental traces, which was selected randomly. The glutamate concentration-response curve (glutamate concentration are 0.03, 0.1, 0.3, 1, 3, and 100 µM and glycine concentrations are 10 and 30 µM) and the glycine concentration-response curve (glycine concentrations are 0.01, 0.03, 0.1, 0.3, 1, 30 µM, and glutamate concentration are 30 and 100 µM). Each curve was normalized to the maximal response from each oocyte in the absence of **EU1794-4**. Data are from 4 to 12 paired oocytes recordings from at least two independent experiments.

DOI: https://doi.org/10.7554/eLife.34711.007

*Figure 3 continued on next page*

*Figure 3 continued*

The following figure supplement is available for figure 3:

**Figure supplement 1.** The effects of EU1794-27 and EU1794-4 on agonist potency for all diheteromeric NMDARs.

DOI: https://doi.org/10.7554/eLife.34711.008

EU1794-27 was also determined for GluN1/GluN2A (*Figure 4C,D*). Higher $K_d$ values (lower potency) were determined for each molecule at GluN1/GluN2A as compared to GluN1/GluN2D that parallel $EC_{50}$ values determined using *X. laevis* oocytes. Similar to oocyte data, robust inhibition was produced by **EU1794-2**, modest inhibition was produced by **EU1794-4**, and transient inhibition and recovery was observed by the application of **EU1794-27**. Interestingly, the extent of steady-state modulation for this series appeared to be dependent on the level of desensitization of the receptors at the time of modulator application. When classifying the cells as either having high or low levels of desensitization (using 35% steady-state/peak response as a cut-off), the extent of modulation by 10 μM was significantly different for **EU1794-27** (p=0.02, unpaired t-test, N = 4,2, respectively). The low desensitized group (40% average desensitization) was modulated by 107%, whereas the high desensitized group (10% average desensitization) was modulated to 247% of control by 10 μM **EU1794-27** (*Figure 4C* and *Figure 4—figure supplement 1F*).

Given that other NMDAR modulators have displayed agonist-dependence (*Petrovic et al., 2005*; *Acker et al., 2011*; *Hansen and Traynelis, 2011*; *Borovska et al., 2012*; *Vyklicky et al., 2015*; *Wang et al., 2017*), we performed rapid solution exchange experiments to examine if these modulators had different affinities at agonist-bound and *apo* receptors (*Figure 5A,B*). The instantaneous current response to a rapid step into glutamate plus glycine (from glycine alone) gives an estimate of whether a modulator was pre-bound to receptors with only glycine (but not glutamate) bound. The immediate NMDAR activation after a rapid switch to a solution containing glutamate and glycine should occur faster than modulator binding. We chose to test for agonist-dependence using GluN1/GluN2D since they lack desensitization, which would complicate interpretation of this modulator property. We found that the peak current was similar when cells were preincubated in glycine with or without 3 μM **EU1794-2**, suggesting that **EU1794-2** does not bind appreciably to the receptor in the absence of glutamate (*Figure 5A,C*). After the rapid step into glutamate where glycine and modulator were pre-exposed, we observed a relaxation to a new response level that was similar in amplitude to that observed with steady state co-application of glutamate plus glycine and modulator (*Figure 5A,C*). Interestingly, we also observed a similar effect for the converse experiment, in which we pre-applied glutamate with or without **EU1794-2**, followed by a rapid step into glutamate plus

**Table 3.** EU1794-27 effects on glutamate and glycine $EC_{50}$ values.

$EC_{50}$ (μM) [conf. int.]*

| | Control Glutamate | EU1794-27 Glutamate | Fold Difference | Control Glycine | EU1794-27 Glycine | Fold Difference |
|---|---|---|---|---|---|---|
| GluN2A | 2.9 [2.2, 4.0] | 1.1 [0.88, 2.1] | 2.3[†] | 0.57 [0.34, 0.96] | 0.38 [0.23, 0.64] | 1.6 |
| GluN2B | 1.2 [1.1, 1.3] | 0.65 [0.49, 0.85] | 2.0[†] | 0.28 [0.22, 0.36] | 0.14 [0.08, 0.24] | 2.9[†] |
| GluN2C | 0.67 [0.61, 0.73] | 0.22 [0.16, 0.31] | 3.3[†] | 0.23 [0.13, 0.40] | 0.10 [0.07, 0.15] | 2.4 |
| GluN2D | 0.21 [0.17, 0.27] | 0.054 [0.028, 0.11] | 5.3[†] | 0.086 [0.077, 0.095] | 0.020 [0.013, 0.029] | 4.9[†] |

*Glutamate $EC_{50}$ values (in the presence of 30 μM glycine) and glycine $EC_{50}$ values (in the presence of 100 μM glutamate) were obtained by least-squares fitting of data from independent oocyte recordings by the Hill equation. $EC_{50}$ values are given as the mean with the 95% confidence interval determined from log($EC_{50}$). Data in the absence and presence of **EU1794-27** were obtained from the same oocyte. Data are from 5 to 12 paired oocytes recordings from at least two independent experiments.

[†]indicates paired measurements with non-overlapping confidence intervals.

DOI: https://doi.org/10.7554/eLife.34711.010

**Table 4.** EU1794-4 effects on glutamate and glycine EC$_{50}$ values.

| | EC$_{50}$ ($\mu$M) [conf. int.]* | | | | | |
| --- | --- | --- | --- | --- | --- | --- |
| | Control Glutamate | EU1794-4 Glutamate | Fold Difference | Control Glycine | EU1794-4 Glycine | Fold Difference |
| GluN2A | 3.6 [3.3, 4.0] | 1.5 [1.2, 2.1] | 2.4[†] | 0.89 [0.72, 1.1] | 0.35 [0.17, 0.70] | 2.8[†] |
| GluN2B | 1.5 [1.2, 2.0] | 0.8 [0.48, 1.3] | 2.0[†] | 0.46 [0.37, 0.56] | 0.14 [0.12, 0.17] | 3.3[†] |
| GluN2C | 0.84 [0.57, 1.2] | 0.29 [0.15, 0.55] | 3.2[†] | 0.31 [0.15, 0.62] | 0.08 [0.024, 0.25] | 4.6[†] |
| GluN2D | 0.63 [0.41, 0.95] | 0.17 [0.065, 0.43] | 3.9[†] | 0.15 [0.089, 0.24] | 0.016 [0.007, 0.038] | 8.9[†] |

*Glutamate EC$_{50}$ values (in the presence of 10 $\mu$M glycine) and glycine EC$_{50}$ values (in the presence of 30 $\mu$M glutamate) were obtained by least-squares fitting of data by the Hill equation. EC$_{50}$ values are given as the mean with the 95% confidence interval determined from log(EC$_{50}$). Data in the absence and presence of EU1794-4 were obtained from the same oocyte. Data are from 4 to 8 paired oocytes recordings from two independent experiments.
[†]indicates paired measurements with non-overlapping confidence intervals.

DOI: https://doi.org/10.7554/eLife.34711.011

glycine and **EU1794-2** (*Figure 5B,C*). We interpret these data to suggest that **EU1794-2** associates with the receptor with higher affinity after glutamate and glycine binding.

We repeated these use-dependent experiments with **EU1794-27**, which yielded a similar result, although a slightly different experimental design was required for consistent responses. The pre-application of glycine with **EU1794-27** produced a similar level of immediate activation following glutamate application (*Figure 5D,E*). This was then followed by the complex actions observed when **EU1794-27** was applied to steady-state responses of NMDARs (*Figure 5D*, *right panel*). Using the same protocol to pre-apply glutamate became problematic for some cells due the degree to which **EU1794-27** enhances agonist potency, a feature exemplified by the prolongation of deactivation. This action of **EU1794-27** was able to render nanomolar contaminate levels of glycine (when present) more active, which necessitated a different experimental design (*Figure 5—figure supplement 1A*). When **EU1794-2** was used in this experimental paradigm, inhibition of the contaminate level of activity was observed upon modulator application (*Figure 5—figure supplement 1B*). Nevertheless, we still observed a similar result with pre-application of glycine and **EU1794-27**, with the instantaneous response reaching the control level, followed by a slow relaxation to a new potentiated level that reflected the time course for association of **EU1794-27** after binding of both glutamate and

**Table 5.** Comparison of **EU1794-27**, **EU1794-4**, and **EU1794-2** effects at GluN1/GluN2C NMDAR responses to sub-saturating agonist.

| GluN2C | Glu/Gly ($\mu$M) | Relative agonist Response* (% of maximal response) | Modulation by 10 $\mu$M[†] (% of control) |
| --- | --- | --- | --- |
| EU1794-27 | 100/30 1/0.3 | - 20 ± 1.6 | 190 ± 10% 480 ± 31%[‡] |
| EU1794-4 | 100/30 1/0.3 0.3/0.09 | - 27.7 ± 3.3 2.8 ± 0.7 | 54 ± 1.6% 140 ± 12%[‡] 660 ± 110%[‡] |
| EU1794-2 | 100/30 0.6/0.2 | - 21.5 ± 6.8 | 16 ± 1.0% 43 ± 2.1%[‡] |

The data presented are mean ±SEM.
*Reported average responses to sub-saturating agonist were normalized to a 100/30 $\mu$M glutamate/glycine response.
[†]The degree modulation is reported as a percent of the control response to sub-saturating agonist. Some of these data points correspond to those used to calculated EC$_{50}$ values for each modulator in **Table 1**, **Table 2** or **Supplementary file 1**. Data are mean ±SEM.
[‡]$p < 0.05$ as compared to 100/30 $\mu$M glutamate/glycine response, determined by an unpaired t-test, n = 4–10 cells.

DOI: https://doi.org/10.7554/eLife.34711.012

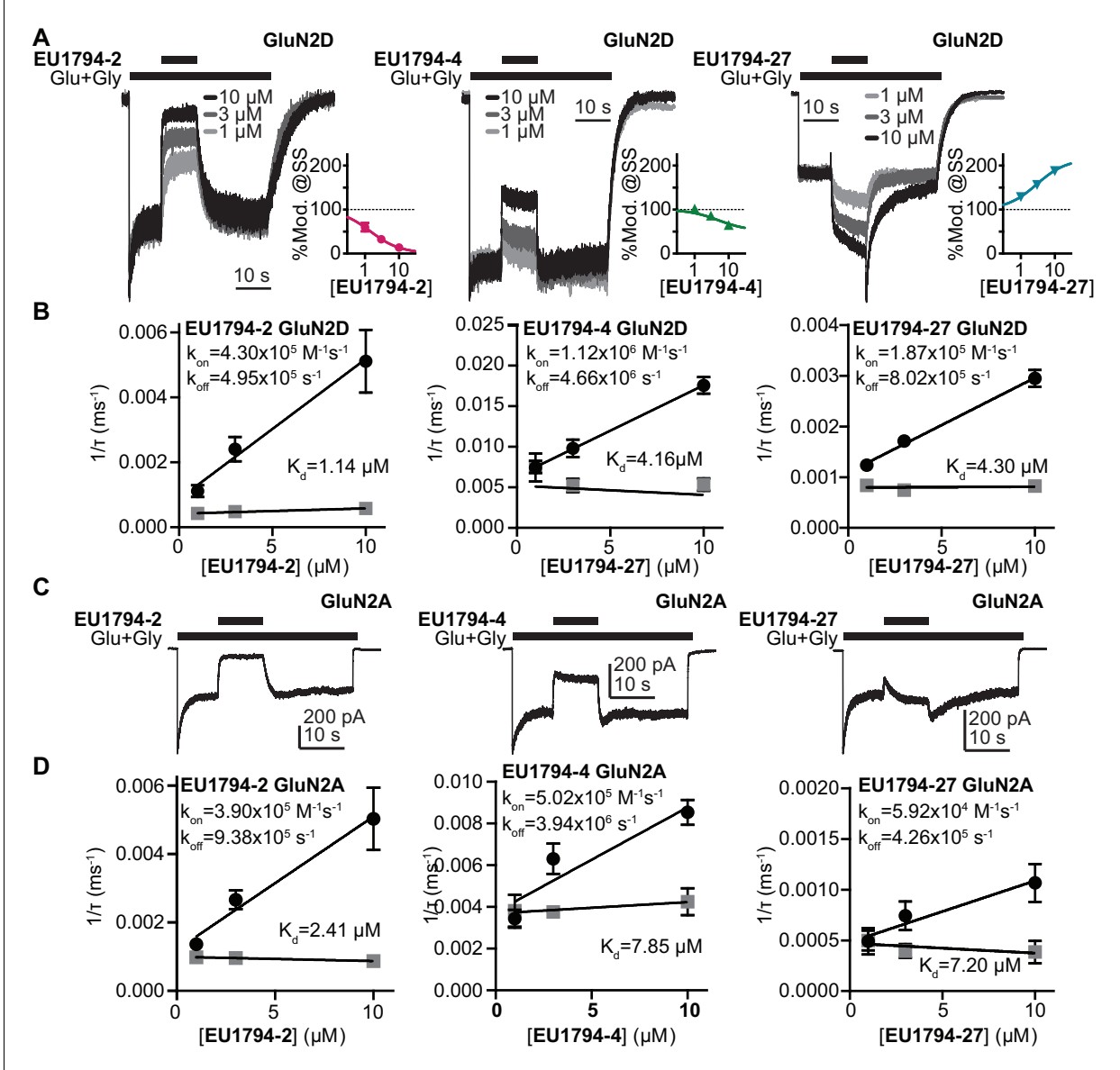

**Figure 4.** The association and dissociation time course of **EU1794-2**, **EU1794-4** and **EU1794-27** differ at GluN1/GluN2A and GluN1/GluN2D. (A) Representative whole-cell current recordings of GluN1/GluN2D NMDARs exemplifying the modulation time course of **EU1794-2**, **EU1794-4** and **EU1794-27** (1, 3, 10 μM). Responses were evoked by 100/30 μM glutamate/glycine and were normalized for display. Insets show the steady state modulation extent from the HEK cell experiments fit by the Hill equation (Hill slope fixed to 1). (B) Analysis of the time course for the onset of PAM and NAM action as a function of concentrations for **EU1794-2** (*left*), **EU1794-4** (*left*) and **EU1794-27** (*right*) used to determine the association rate $k_{ON}$ (black circles) and dissociation rate $k_{OFF}$ (grey squares), from which we can calculate $K_d$ for each modulator. Data are from at least 3 cells for each concentration of modulator. For **EU1794-27** the association and dissociation rate linear component was also found to be concentration dependent ($m_{on} = -0.98$ pA/(ms*μM), $m_{off} = 0.62$ pA/(ms*μM)). (C,D) Similar set of panels as A,B but examining **EU1794-2**, **EU1794-4** and **EU1794-27** modulation of GluN1/GluN2A. In example recordings, all modulator concentrations are 10 μM. Data are from at least 3 cells for each concentration of modulator. For **EU1794-27** time-course was fitted with the sum of an exponential and linear function, and both the time constant and slope were concentration dependent ($m_{on} = -0.33$ pA/(s*μM), $m_{off} = 0.16$ pA/(s*μM)).

DOI: https://doi.org/10.7554/eLife.34711.013

The following figure supplement is available for figure 4:

**Figure supplement 1.** The actions of **EU1794** modulators are voltage-independent and are influenced by the desensitization level.

DOI: https://doi.org/10.7554/eLife.34711.014

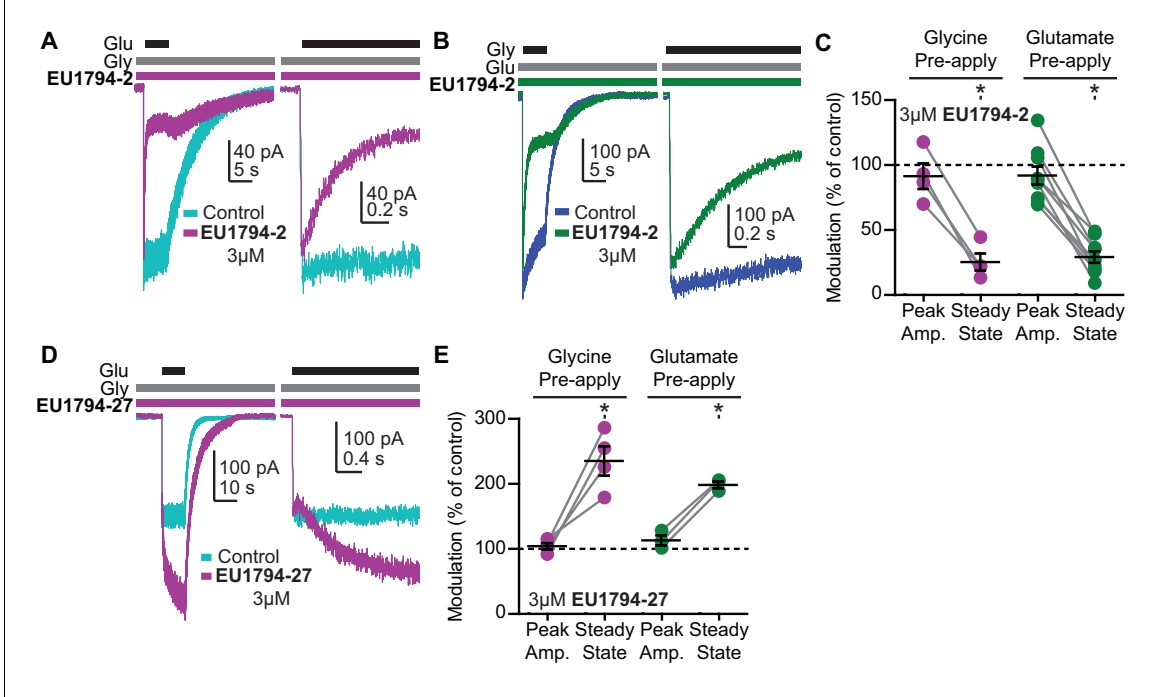

**Figure 5.** The EU1794 series has agonist-dependence that alters the response time-course. (A) Representative whole-cell recordings of GluN1/GluN2D NMDARs exemplifying the **EU1794-2** dependence on glutamate binding. The cell was exposed continuously to 30 µM glycine with/without 3 µM **EU1794-2**, and then was rapidly stepped into a solution additionally containing 100 µM glutamate. The *right panel* shows an expanded view of the rise of the receptor response and the rapid time course of **EU1794-2** action. (B) Representative whole-cell current recordings of GluN1/GluN2D NMDARs exemplifying the **EU1794-2** dependence on glycine binding. The cell was exposed continuously to 100 µM glutamate with/without 3 µM **EU1794-2**, and then was rapidly stepped into a solution additionally containing 30 µM glycine. The *right panel* shows an expanded view of the rise of the receptor response and the rapid time course of **EU1794-2** action. (C) Quantification of the modulation of the immediate peak response and steady state data (shown as a percentage EU1794-2/control for each phase) from agonist-dependence experiments for **EU1794-2** (3 µM, *Figure 5A,B* and S4B). * signifies p<0.05 as determined by a Bonferroni's multiple comparison test where only **EU1794-2** *vs* control was compared for both peak and steady state conditions after a significant repeat measures ANOVA (F(3,3) = 7.66 for pre-applied glycine and F(3,7) = 12.4 for pre-applied glutamate). (D) Representative whole-cell current recordings of GluN1/GluN2D NMDARs exemplifying the **EU1794-27** dependence on glutamate binding. The cell was exposed continuously to 30 µM glycine with/without 3 µM **EU1794-27,** and the solution was rapidly changed to one additionally containing 100 µM glutamate. The *right panel* is an expanded view of the activation kinetics of the responses and the onset of modulation. (E) Quantification of the modulation of the immediate peak and steady state response (shown as a percentage EU1794-27/control for each phase) from agonist-dependence experiments for **EU1794-27** (3 µM, *Figure 5D* and S4A). * signifies p<0.05 from the *post hoc* Bonferroni's multiple comparison test (F(3,3) = 17.8 for pre-applied glycine and F(3,2) = 13.5 for pre-applied glutamate).

DOI: https://doi.org/10.7554/eLife.34711.015

The following figure supplement is available for figure 5:

**Figure supplement 1.** EU1794 modulators are dependent on co-agonist binding.

DOI: https://doi.org/10.7554/eLife.34711.016

glycine (*Figure 5E*). This result illustrates a requirement for glycine to be bound to the receptor for high-affinity binding of **EU1794-27**. To circumvent any ambiguity associated with glycine contamination, we utilized another experimental design described by *Vyklicky et al. (2015)*. Glutamate was pre-applied with 7-CKA (100 µM) to antagonize the glycine site, blocking any occupancy by contaminant glycine. The receptor was activated by switching to a solution that lacked 7-CKA and contained glycine plus glutamate (with or without modulator, *Figure 5—figure supplement 1C*). To test the ability of **EU1794-27** to bind to the receptor with the GluN1 ABD bound to antagonist, **EU1794-27** (3 µM) was added to the solution containing 7-CKA. Upon the switch from the 7-CKA/**EU1794-27** solution containing glutamate to a solution just containing glutamate and glycine, no detectable change in the response rise time or peak amplitude was observed. If **EU1794-27** bound during the 7-CKA phase prior to agonist binding, we would have expected an increased instantaneous peak current upon switching to glutamate plus glycine given that the dissociation of **EU1794-27** is slower

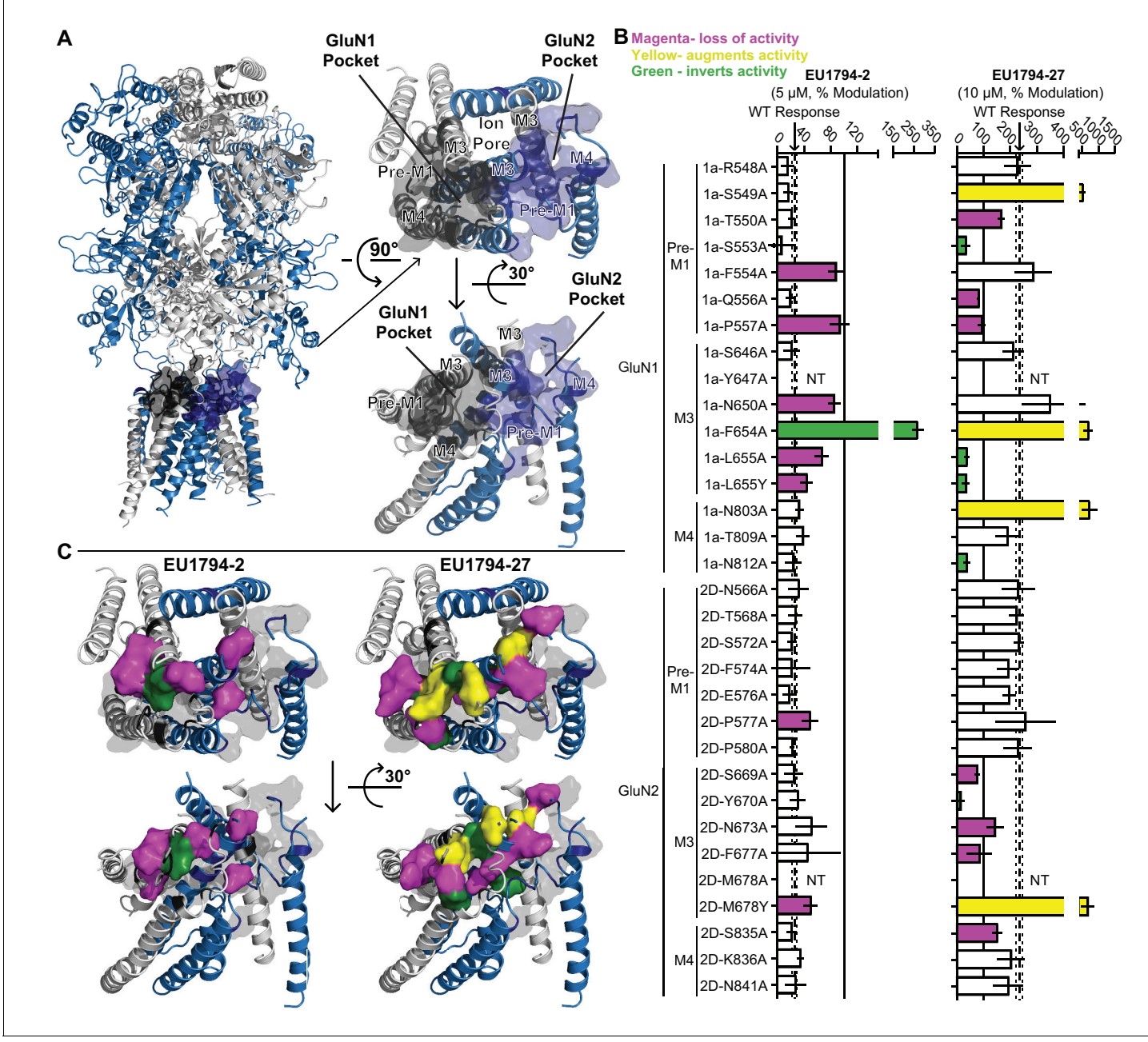

**Figure 6.** Distinct and overlapping pattern of residues contribute to the actions of **EU1794** modulators with opposing effects. (**A**) Ribbon representation (*left*) of a GluN1/GluN2D model based on published crystal structures of GluN1/GluN2B and GluA2 (*Karakas and Furukawa, 2014*; *Lee et al., 2014*; *Yelshanskaya et al., 2016*). The GluN1 subunits are grey and the GluN2D are purple; surface shell indicate residues of interest in GluN1 and GluN2D, including residues that are homologous to those identified in recent studies investigating transmembrane domain interacting modulators of NMDAR and AMPAR (*Yelshanskaya et al., 2016*; *Wang et al., 2017*). These GluN1 and GluN2D residues were changed to alanine (or tyrosine in two cases, as previously reported) and tested for effects on the NMDAR sensitivity to **EU1794-2** and **EU1794-27**. Top down (*right top*) and side (*right bottom*) view of the model TMD and linker segments highlight two potential pockets with GluN1 and GluN2 M3 helices in the background. One pocket is primarily associated with GluN1 and the other GluN2. (**B**) The responses (mean ±99% CI) in **EU1794-2** (5 µM, *left*) and **EU1794-27** (10 µM, *right*) are shown as a % of the mutant receptor response to 100 µM glutamate and 30 µM glycine in the absence of the test compound. The wild type mean response is shown by the dashed line, surrounding dotted lines, which indicate the 99% confidence interval. Residues with non-overlapping confidence intervals with the wild type are colored with magenta indicating apparent loss of activity, yellow indicating augmented activity, and green indicating robust inverted modulation. Data for the mutations shown represent 4–18 oocytes from at least two independent experiments and the wild type response shown represent 83 oocytes for **EU1794-2** and 76 oocytes for **EU1794-27**, recorded each experimental day to ensure consistency. We were not able to test all mutant receptors due to low expression for some (NT, not tested). (**C**) The residues (shown by the space-filling shell) identified in (**B**) that altered the

*Figure 6 continued on next page*

*Figure 6 continued*

effects of **EU1794-2** (*left*) and/or **EU1794-27** (*right*) were mapped onto the model with colors as described in (**B**). Views are the same as (**A**), with the all residues probed shown as grey transparent shell representation.

DOI: https://doi.org/10.7554/eLife.34711.017

The following figure supplements are available for figure 6:

**Figure supplement 1.** EU1794 series effects are not altered by ATD perturbations.

DOI: https://doi.org/10.7554/eLife.34711.018

**Figure supplement 2.** Compound competition screen of the **EU1794** series with NMDAR modulators highlights potential interactions with the ABD and the TMD.

DOI: https://doi.org/10.7554/eLife.34711.019

**Figure supplement 3.** Residues in known modulator binding sites do not perturb actions of the **EU1794** series.

DOI: https://doi.org/10.7554/eLife.34711.020

**Figure supplement 4.** Mutated residues that impact **EU1794** actions have lower open probability.

DOI: https://doi.org/10.7554/eLife.34711.021

than 7-CKA (compare the steady state response and the 7-CKA response in *Figure 5—figure supplement 1C*). Additionally, comparable experiments were performed with 100 µM APV with similar results, as APV stabilizes the GluN2 agonist binding domain (ABD) open-cleft conformation, and thereby prevents the pre-binding of **EU1794-27** (*Figure 5—figure supplement 1D*). When **EU1794-27** (3 µM) was included in all solutions, potentiation was observed but APV unbinding was a required prior to modulation (*Figure 5—figure supplement 1D*). Together these data suggest that the **EU1794** series is capable of high affinity binding only when glutamate and glycine are bound to the receptor.

## Evaluation of interactions with known modulatory sites

To identify the molecular determinants of action for the **EU1794** series, we evaluated the ability of GluN2 ATD deletion, GluN1 ATD splice variants, and co-application of known modulators to alter the effects of the **EU1794** modulators. The ATD harbors the binding site for the GluN2B-selective negative allosteric modulator ifenprodil. Deletion of the ATD from GluN2A, GluN2B, or GluN2C had no effect on the actions of the PAM **EU1794-27**; deletion of the ATD from GluN2D reduced but did not eliminate potentiation (*Figure 6—figure supplement 1A*). Similarly, inclusion of 21 residues in the ATD encoded by alternatively spliced GluN1 exon5 only slightly altered the extent of potentiation of **EU1794-27**, but was without effect on $EC_{50}$ for potentiation of GluN2B- and GluN2D-containing NMDARs activated by saturating agonist (*Figure 6—figure supplement 1B*). We previously described a similar result for the negative allosteric modulator **EU1794-2** (*Katzman et al., 2015*). These data are consistent with minimal involvement of the GluN1 or GluN2 ATD in the actions of **EU1794** modulators.

We next screened for interaction with known modulators to focus our search for the molecular determinants of **EU1794** series modulation (*Mullasseril et al., 2010*; *Acker et al., 2011*; *Hansen and Traynelis, 2011*; *Hansen et al., 2012*; *Khatri et al., 2014*; *Ogden et al., 2014*; *Hackos et al., 2016*; *Tajima et al., 2016*; *Yi et al., 2016*) (*Figure 6—figure supplement 2A*). In these experiments, a known positive or negative modulator was co-applied with either **EU1794-2** or **EU1794-27**, with each pair always containing one PAM and one NAM. If there is no interaction between paired modulators, their combined activity should be predicted by multiplying the extent of their independent actions. Co-application of the modulator pairs ifenprodil/**EU1794-27**, **EU1794-2**/CIQ and **EU1794-2**/PYD-106 produced levels of modulation that largely could be predicted from their independent actions (*Figure 6—figure supplement 2B*, *top row*). Modest differences from predictions were observed with modulators that bind to the ABD interface, TCN-201/**EU1794-27**, **EU1794-2**/GNE-6901 and **EU1794-2**/GNE-0723 (*Figure 6—figure supplement 2B*, *bottom left*). Co-application of the GluN2C/GluN2D-selective negative allosteric modulators QNZ46 and DQP-1105 paired with **EU1794-27** resulted in the greatest divergences from predictions, raising the possibility that these modulators have partially overlapping binding sites (*Figure 6—figure supplement 2B*, *bottom right*) or similar downstream mechanisms.

We also examined the ability of **EU1794-2** and **EU1794-27** to modulate NMDARs harboring mutations within the structural determinants for other known allosteric modulators. Mutations in

GluN1 (I519A, R755A) and GluN2A (L780A, G786A) that block TCN-201 inhibition were evaluated for effects on **EU1794** series of modulators (*Hansen et al., 2012*). NMDARs that contained GluN1/GluN2A(L780A), GluN1/GluN2A(G786A), GluN1(R755A)/GluN2D, GluN1(I519A)/GluN2D were equally sensitive as wild type receptors to inhibition by **EU1794-2** or potentiation by **EU1794-27** (*Figure 6—figure supplement 3A*). GluN1/GluN2A(E530A), GluN1/GluN2A(V783W), GluN1 (Y535W)/GluN2A, GluN1(Y535V)/GluN2D and GluN1(Y535W)/GluN2D, which reduce the actions of GNE-6901 and GNE-0723 (*Hackos et al., 2016*), produced no significant effects on inhibition by **EU1794-2** or potentiation by **EU1794-27** (*Figure 6—figure supplement 3B*). Inhibition by EU1794-2 of GluN1/GluN2C(K470G) and GluN1/GluN2C(S472T), which block PYD-106 potentiation of GluN1/GluN2C (*Khatri et al., 2014*), was similar to wild type NMDARs (*Figure 6—figure supplement 3C*). Interestingly, inhibition by **EU1794-2** and potentiation by **EU1794-27** was not significantly changed by GluN1/GluN2D(Q701Y) and GluN1/GluN2D(L705F), mutations that appear to confer subunit selectivity for GluN2C/D over GluN2A/B for QNZ-46 and DQP-1105 (*Figure 6—figure supplement 3D*) (*Acker et al., 2011*; *Hansen and Traynelis, 2011*).

## Mutagenesis suggests shared structural determinants of action for PAMs and NAMs

The result obtained with the QNZ-46/DQP-1105 interaction test suggested that the two residues we evaluated for QNZ-46 and DQP-1105 insufficiently probed the structural determinants of action for these compounds. We therefore examined a GluA2 AMPA receptor structure bound to CP-465,022, which shares a core scaffold with QNZ modulators (*Yelshanskaya et al., 2016*). Residues identified as being important for CP-465,022 binding in *Yelshanskaya et al., 2016* were aligned to the GluN1 and GluN2D subunits to map this modulatory site onto the NMDAR subunits, in addition to critical residues for GNE-9278 in this same region (*Figure 6A*, *Wang et al., 2017*). This broader range of residues were located on the pre-M1, M3, and pre-M4 regions of both GluN1 and GluN2D, which have previously been suggested to cooperate to control gating (*Ogden and Traynelis, 2013*; *Alsaloum et al., 2016*; *Chen et al., 2017*; *Ogden et al., 2017*; *Yelshanskaya et al., 2017*). These residues were suggested by *Yelshanskaya et al., 2016* to constitute a binding site in homomeric GluA2 AMPARs. However, mapping the homologous residues onto NMDAR structures yields two pockets given the multimeric subunit architecture, one of which consists of the residues of GluN1 and the other of residues of GluN2. Certain residues of M3, depending on their position on the helix, could point towards either pocket, rendering the two pockets to be lined by a mixture of GluN1 and GluN2 residues. 15 GluN1 and 15 GluN2D residues in these two regions that probed these pockets were identified and mutated to allow a test of the contribution of each residue to **EU1794-2** inhibition and **EU1794-27** potentiation. Inhibition by **EU1794-2** was significantly altered by substitutions at six residues in GluN1 and 2 GluN2D residues (*Figure 6B*), which were found on pre-M1 and M3 regions of both GluN1 and GluN2D. Potentiation by **EU1794-27** was more labile, being altered in mutations at 9 GluN1 residues and 6 GluN2D residues (*Figure 6B*). Residues that perturbed the actions of **EU1794-27** were spread across all regions tested except for the GluN2D pre-M1. Modulation was observed to be altered in three different ways by the mutations studied here: activity could be reduced, increased, or inverted.

The distributions of residues that altered modulation by **EU1794-2** and **EU1794-27** shows clear overlap (*Figure 6C*). Inhibition by **EU1794-2** was altered primarily by mutations in GluN1, whereas potentiation by **EU1794-27** was perturbed by residues both in GluN1 and in the M3 helix of GluN2D. Interestingly, the mutations that invert activity of **EU1794-2** and **EU1794-27** were distinct, but in some cases were in close proximity. For example, **EU1794-2** was inverted by GluN1-F654A, whereas mutation of the adjacent residue GluN1-L655A/Y inverted the modulatory action of **EU1794-27** (*Figure 6D*). **EU1794-27** potentiation was also converted to inhibition by four mutations at residues residing on the 3 areas of GluN1 that were investigated (pre-M1, M1, M3) and also on the M3 GluN2D helix, which were in close proximity to each other (*Figure 6D*). **EU1794-27** actions on GluN1-L655A/Y and GluN2D-M678Y, which are homologous residues immediately downstream of the SYTANLAAF motif, resulted in opposite effects (GluN1-L655A/Y converts **EU1794-27** to an inhibitor and GluN2D-M678Y increases the potentiation of **EU1794-27**). We interpret these results to suggest that the activity of both PAMs and NAMs of the **EU1794** series is dependent on multiple residues in the GluN1 subunit, some of which are overlapping. Furthermore, potentiation by **EU1794-27** is dependent on a wider range of GluN1 and GluN2 residues. One possible way to

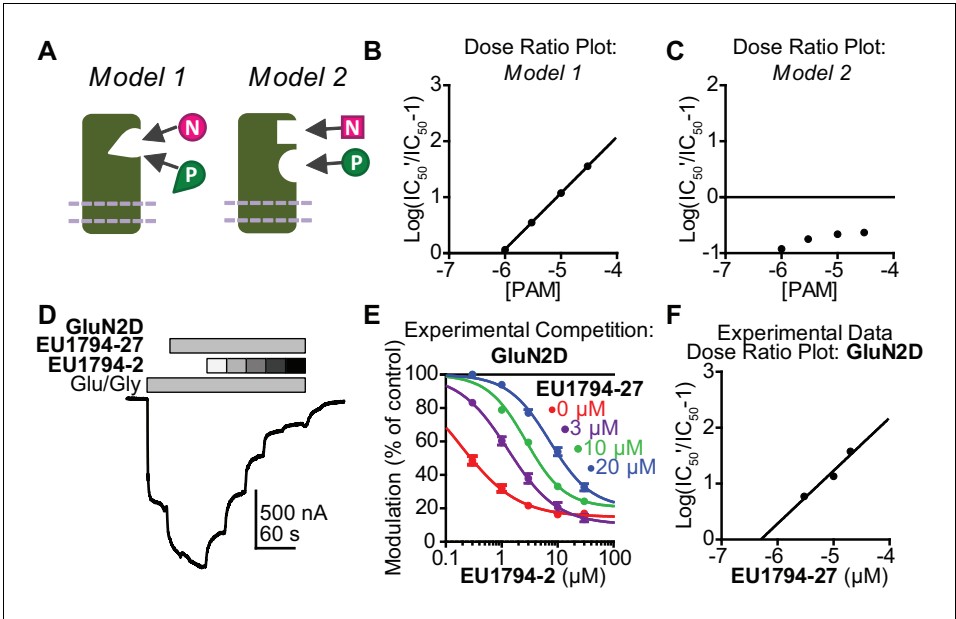

**Figure 7.** EU1794 PAMs and NAMs act in a competitive manner that matches receptor models with mutually exclusive modulator binding. (**A**) Cartoon models show two potential modulator binding schemes. Model 1 possesses mutually exclusive modulator binding whereas Model 2 can bind either or both modulators. Ligands are denoted by [N] (NAM) and [P] (PAM). (**B,C**) Dose-ratio plots are shown for simulated concentration-response curves of the NAM in various concentration of the PAM for *Model 1* (**B**) and *Model 2* (**C**). $IC_{50}'$ is the observed NAM $IC_{50}$ in the presence of a fixed concentration PAM. (**D**) A representative TEVC recording (fixed 3 μM **EU1794-27** and 0.3, 1, 3, 10, 30 μM **EU1794-2**) is shown for GluN1/GluN2D NMDARs (top). (**E**) The mean steady state responses in (**D**) were fitted with the Hill equation. Concentration-response data in the absence of **EU1794-27** are included for comparison from *Figure 1C*. (**F**) Dose-ratio analysis of experimental data illustrates the shift in the $IC_{50}$ of the NAM as a function of PAM concentration. Data are from at least six oocytes evaluated from two independent experiments.

DOI: https://doi.org/10.7554/eLife.34711.022

The following figure supplement is available for figure 7:

**Figure supplement 1.** EU1794 PAMs and NAMs act in a competitive manner that suggests mutually exclusive modulator binding.

DOI: https://doi.org/10.7554/eLife.34711.023

account for this would be if both modulators bound near the GluN1 pre-M1 helix, with potentiator actions dependent on the nearby GluN2 M3 residues associated with the GluN1 pocket (*Figure 6A, C*).

To assess whether mutated residues impacted **EU1794** series modulation through altering receptor open probability, we estimated mutant open probability using MTSEA modification of an introduced cysteine residue (*Jones et al., 2002*; *Yuan et al., 2005*). The open probability of GluN2D-containing receptors is very low and may not allow for precise determination of the effects of mutations, thus we used GluN2B-containing receptors for comparison. GluN1 mutations were expressed with a GluN2B A7C construct (See *Figure 6—figure supplement 4* legend) and GluN2 mutations were expressed with a GluN1 A7C construct. Several residues were assayed that resulted in reversed, enhanced, or blocked modulator activity (*Figure 6—figure supplement 4A*). GluN1-P557A, which blocked activity of both **EU1794-2** and **EU1794-27**, had an increased open probability as compared to wild type. GluN1-F654A, which converted **EU1794-2** into a potentiator and enhanced **EU1794-27** potentiation, had a similar open probability as compared to wild type (*Figure 6—figure supplement 4A*). Interestingly, mutation of the homologous GluN2 residue (GluN2-F653A), which reduced **EU1794-27** potentiation, also lowered open probability as compared to wild type (*Figure 6—figure supplement 4B*). *Figure 6—figure supplement 4* shows that GluN1-L665A and GluN1-N803A, which both converted **EU1794-27** into an inhibitor, had significantly lower open

**Table 6.** The response to co-application of **EU1794-2** and **EU1794-27** suggest a common binding site.

| | EU1794-2 IC$_{50}$ (μM) [conf. int.]* | | | | EU1794-27 EC$_{50}$ (μM) [conf. int.]* | | |
| --- | --- | --- | --- | --- | --- | --- | --- |
| | +EU1794-27 0 μM | +EU1794-27 3 μM | +EU1794-27 10 μM | +EU1794-27 20 μM | +EU1794-2 0 μM | +EU1794-2 1 μM | +EU1794-2 3 μM |
| GluN2A | 0.60[†] [0.44, 0.82] | 2.5[‡] [1.9, 3.1] | 3.6[‡] [2.4, 4.8] | 8.0[‡] [1.3, 15] | _[§] | _[§] | _[§] |
| GluN2B | 1.2[†] [0.8, 1.9] | NR | 3.5[‡] [2.1, 4.9] | 12 [~, 26] | 1.4[†] [1.2, 1.7] | 6.0[‡] [4.6, 7.4] | 8.4 [~, 18] |
| GluN2C | 0.21[†] [0.18, 0.25] | 1.2[‡] [0.91, 1.4] | 2.4[‡] [1.8, 3.0] | 6.0[‡] [4.1, 7.8] | 2.8[†] [2.5, 3.2] | 7.3[‡] [5.9, 8.8] | 7.3 [2.1, 11.74] |
| GluN2D | 0.20[†] [0.17, 0.25] | 1.4[‡] [1.2, 1.6] | 2.7[‡] [2.3, 3.2] | 7.3[‡*] [4.6, 9.9] | 2.4[†] [1.8, 3.3] | 7.5[‡] [5.6, 9.3] | 7.8 [1.1, 14] |

A five point concentration response curve was obtained for modulator effects on responses to maximally effective concentrations of glutamate and glycine (100/30 μM) for **EU1794-2** co-administered with increasing concentrations of **EU1794-27**, and for **EU1794-27** in increasing concentrations of **EU1794-2**.

*IC$_{50}$ and EC$_{50}$ values were obtained by least-squares fitting of data by the Hill equation. IC$_{50}$ and EC$_{50}$ values are given as the mean with the 95% confidence interval determined from log(IC$_{50}$) or log(EC$_{50}$). Data are from at least six oocytes evaluated from two independent experiments for the **EU1794-2** IC$_{50}$ determinations and from at least three oocytes evaluated from one experiment for the **EU1794-27** EC$_{50}$ determinations.

NR, Given the low potency for modulators at GluN2B, the **EU1794-2** concentration-response curve with 3 μM **EU1794-27** determination was not recorded.

[†] Data from **Table 1** is included here for clarity.

[‡]indicates non-overlapping 95% confidence interval with the **EU1794-2** IC$_{50}$ without **EU1794-27** or **EU1794-27** EC$_{50}$ without **EU1794-2**.

[§]**EU1794-27** does not potentiate GluN2A responses to maximally effective concentrations of glutamate and glycine.

~indicates that the confidence interval reached the theoretical limit (EC$_{50}$ = 0).

DOI: https://doi.org/10.7554/eLife.34711.024

probabilities as compared to wild type NMDARs (**Figure 6—figure supplement 4A**). These data suggest that altered **EU1794** modulator action by mutated residues is not directly dependent on alterations in open probability.

## PAMs and NAMs exert their opposing effects via a shared binding site on NMDARs

Given the similarity in chemical structure between positive and negative modulators in this series, we hypothesized that they might have overlapping binding sites. In order to conduct a detailed

**Table 7.** Enantiomeric preference of **EU1794-27**.

| | EC$_{50}$ (μM) [conf. int.]* Maximal Modulation Extent (% of control) | | | |
| --- | --- | --- | --- | --- |
| Compound | EU1794-27 | (-)-EU1794-27 | (+)-EU1794-27 | |
| GluN2A | 7.4 [5.3, 10][†] 52 ± 7% | 5.6 [2.7, 12] 67 ± 5% | - | [‡] |
| GluN2B | 1.4 [1.2, 1.7][†] 130 ± 3% | 2.8 [2.0, 3.8] 170 ± 10% | - | [‡] |
| GluN2C | 2.8 [2.5, 3.2][†] 230 ± 10% | 5.1 [4.1, 6.4] 260 ± 24% | 7.1 [3.6, 14] 160 ± 6.2% | [‡] |
| GluN2D | 2.4 [1.8, 3.3][†] 250 ± 8% | 8.2 [6.7, 9.9] 340 ± 21% | 7.4 [3.7, 15] 130 ± 4.7% | [‡] |

*EC$_{50}$ values were obtained by least-squares fitting of data from individual experiments by the Hill equation. EC$_{50}$ values are given as the mean with the 95% confidence interval determined from log(EC$_{50}$); the maximal degree of modulation is given as mean ±SEM. Data are from 4 to 7 oocytes from at least two independent experiments. Data were not fitted (shown as -) if the response recorded at 30 μM of test compound did not differ by more than 15% from control.

[†]Data from **Table 1** is included here for clarity.

[‡]indicates significant unpaired t-test between the percent modulation between 30 μM (-)-**EU1794-27** and (+)-**EU1794-27**.

DOI: https://doi.org/10.7554/eLife.34711.027

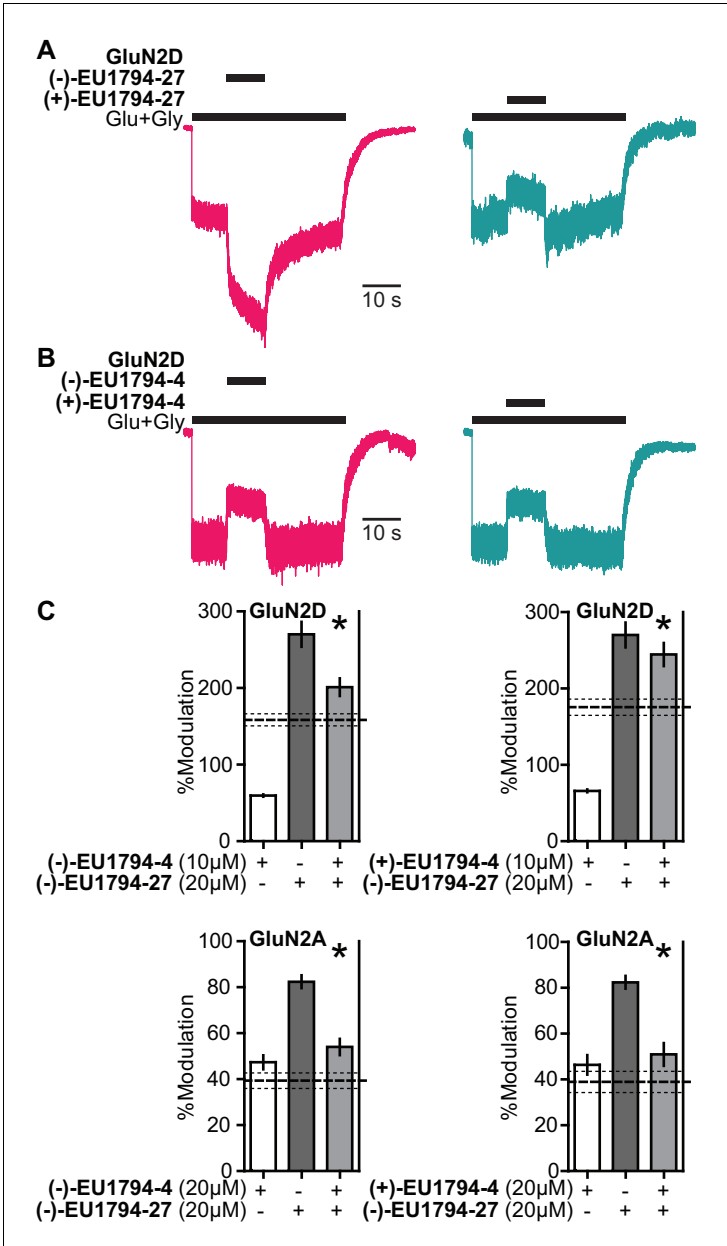

**Figure 8.** The enantiomers of **EU1794-4** and **EU1794-27** have different action. (**A**) Representative modulation of GluN1/GluN2D responses by the enantiomers of **EU1794-27** (both 10 μM, 100/30 μM glutamate/glycine). Similar results were observed in at least 3 cells to fresh modulator solution made less than one hour before experiment conclusion. (**B**) Representative modulation of GluN1/GluN2D responses by the enantiomers of **EU1794-4** (both 10 μM, 100/30 μM glutamate/glycine). Similar results were observed in at least 3 cells to fresh modulator solution made less than one hour before experiment conclusion. (**C**) Single concentration competition experiments illustrating an interaction between both enantiomers of **EU1794-4** (+and -) and the positive modulating enantiomer of **EU1794-27** (-) at GluN1/GluN2D. The individual average effects of the pairs of modulators along with the co-applied effect are plotted, and the predicted mean net effect of the modulators is displayed as a dashed line (SEM, dotted lines, n = 10 from two independent experiments). * signifies p<0.05 as determined by a paired t-test comparing co-applied modulators and predicted net effect of modulator actions.
DOI: https://doi.org/10.7554/eLife.34711.025

The following figure supplement is available for figure 8:

**Figure supplement 1.** The concentration response and stability of the enantiomers of **EU1794-27** and **EU1794-4**.
DOI: https://doi.org/10.7554/eLife.34711.026

functional analysis of competition between allosteric modulators, we first used modeling to examine a receptor's functional response to co-application of positive and negative modulators acting at the same or different sites. We evaluated two models of modulator binding built into a scheme first proposed by (*Lester and Jahr, 1992*). Model 1 describes receptors with a single binding site that can accommodate either PAM or NAM, but not both. Model 2 describes a receptor to which both modulators can bind at the same time in different sites (*Figure 7A*, *Figure 7—figure supplement 1*). We evaluated hypothetical responses from these models (see Materials and methods, *Figure 7—figure supplement 1A–C*). Responses to maximally effective concentrations of agonist were simulated at various concentrations of the positive and negative modulators. When the modulators compete for the same binding site, they act similarly to stated theory about competitive antagonists (*Arunlakshana and Schild, 1959*; *Christopoulos and Kenakin, 2002*), with the PAM causing a rightward shift of the NAM $IC_{50}$ and generating a linear relationship in dose ratio analysis (*Figure 7B*, *Figure 7—figure supplement 1D,E*, *Model 1*). By contrast, there is no apparent $EC_{50}$ shift when the two modulators are capable of binding simultaneously (*Figure 7C*, *Figure 7—figure supplement 1D,E*, *Model 2*). Additional simulations showed the reciprocal effect of NAM on PAM $EC_{50}$ (*Figure 7—figure supplement 1F,G*). We subsequently performed competition experiments to test the hypothesis that this series of PAMs and NAMs compete for a mutually exclusive modulatory pocket (*Figure 7D*). We observed that fixed concentrations of the PAM **EU1794-27** caused parallel shifts in the concentration-response curve of the NAM **EU1794-2** (*Figure 7E*, *Table 6*). A similar phenomenon was shown for the reverse, as fixed concentrations of **EU1794-2** caused parallel shifts in the **EU1794-27** concentration-response curves (*Table 6*). These results closely match *Model 1*, where PAMs and NAMs bind in a mutually exclusive fashion. This suggests that the opposing modulators in the **EU1794** series share overlapping binding sites instead of the coincidence that the subtle chemical differences confer unique binding sites.

The enantiomers of **EU1794-27** were separated to determine whether the racemic activity reflected the actions of only one enantiomer (see Supplemental methods). (-)-**EU1794-27** potentiated NMDAR responses similarly to the racemic mixture (*Figure 8—figure supplement 1A*, *Table 7*). By contrast, (+)-**EU1794-27** exhibited only weak potentiation of GluN1/GluN2C and GluN1/GluN2D, which may be due to the purity achieved via chiral separation (*Table 7*). We previously reported (*Katzman et al., 2015*) that purified enantiomers were likely to racemize in aqueous solutions, which should proceed by a first-order reaction dependent on multiple factors such as ionic strength, pH, buffer, etc. (*Smith et al., 1978*). Thus, to quantitatively determine if racemization would impact our experiments using the enantiomers, we performed a functional assessment of this property in our standard experimental solution. We observed racemization of the enantiomers of **EU1794-27** with a half-life of 197 min (*Figure 8—figure supplement 1B*, *Table 7*). Both enantiomers of **EU1794-4** inhibited NMDAR responses, but with different potencies (*Figure 8—figure supplement 1C*, *Supplementary File 2*). A similar rate of racemization was observed for the enantiomers of **EU1794-4** (half-life of 196 min, *Figure 8—figure supplement 1D*). All enantiomer studies, other than the racemization time-course, were performed rapidly to minimize any racemization, being completed in less than 80 min after making the aqueous solution of modulator. Similar to results in oocytes, when the enantiomers of **EU1794-27** were applied to steady-state GluN1/GluN2D responses in HEK cells, (-)-**EU1794-27** potentiated responses whereas (+)-**EU1794-27** had a slight inhibitory effect (*Figure 8A*). In additional, 10 µM of both enantiomers of **EU1794-4** had similar inhibitory actions on GluN1/GluN2D responses to maximally effective concentrations of glutamate and glycine (*Figure 8B*).

To assess whether both enantiomers bound to similar or distinct sites on the NMDAR, we performed single concentration competition experiments. Co-application of each enantiomer of **EU1794-4** with the (-)-**EU1794-27** resulted in a degree of modulation that was significantly different than that predicted for independent sites of action at both GluN1/GluN2A and GluN1/GluN2D (*Figure 8C*). Interestingly, the competition by **EU1794-4** was not dependent on the direction of modulation by (-)-**EU1794-27**, which had a potentiating action at GluN1/GluN2D (*Figure 8C*, *top panels*) and an inhibitory effect at GluN1/GluN2A (*Figure 8C*, *bottom panels*). Although there remain potential caveats, the evidence suggests that the co-application of (-)-**EU1794-27** with either enantiomer of **EU1794-4** displays mutual exclusivity in modulator binding.

## Discussion

This study highlights how subtle chemical variations in a series of NMDAR allosteric modulators can result in fundamentally different actions on NMDAR responses to maximally effective concentrations of agonists. This series of small molecule allosteric modulators show a spectrum of effects ranging from strong negative modulation to robust positive modulation. In addition, there are features that are shared across this series, including agonist-dependence and the ability to enhance agonist potency. Thorough analysis of the actions of this series illustrates a potential way forward in designing new analogues to achieve a wide range of activities. Additionally, the novel features found in this class of modulators suggest potential new strategies for targeting distinct populations of NMDARs, such as extrasynaptic receptors that typically are not activated by high concentrations of glutamate or distinct patterns of stimulation.

### The site and mechanism of action for the EU1794 series

In the evaluation of the kinetic properties of modulator action, we observed complex actions of **EU1794-4** and **EU1794-27**, which could arise from multiple binding sites, enantiomers of these compounds, or could reflect distinct modulator-dependent mechanistic actions from occupancy of a single binding site. The modulation by **EU1794-4** and **EU1794-27** is voltage independent, eliminating potential channel block within the ion channel pore by potential cationic species as a confounding site that contributes to the observed effects. Given that there is high homology between GluN1 and GluN2, especially in the ABD and the TMD, a reasonable hypothesis is that multiple binding sites exist for **EU1794** series modulators in homologous regions on GluN1 and GluN2 subunits. However, the mutagenesis data argue against this idea, given that the residues at which mutations perturbed the actions of **EU1794-2** and **EU1794-27** overlapped and clustered around the pre-M1, M3, and M4 of the GluN1 subunit. Whereas there were a few residues of the GluN2 M3 helix that influenced **EU1794-27** modulation, the structural NMDAR models suggest that these residues face the GluN1 M3 helix, and thus could interact with the pocket adjacent to the GluN1 pre-M1 helix. Additionally, it's likely the M3 helices of both GluN1 and GluN2, which are in close contact, act in concert with one another to control rapid pore opening or closing. If these residues identified by mutagenesis controlled conformational changes downstream of the **EU1794** binding site(s), it would limit potential binding site candidates to the interface between the GluN1 and the GluN2 ABDs, which would then have only two identical binding sites in diheteromeric NMDARs. Therefore there would be less potential for non-identical binding sites that contribute to the mixed actions that are observed for **EU1794-27** and **EU1794-4**. For these reasons, the idea that the positive and negative allosteric actions reflect modulator specific mechanisms from occupancy of a single site seems the most plausible interpretation.

Evaluation of the effects of enantiomers provides further insight into the binding site of the **EU1794** series. **EU1794-27** exhibits strong stereoselectivity, with the (-) enantiomer showing a typical potentiation time course and the (+) enantiomer producing inhibition. The enantiomers of **EU1794-4** are both capable of producing inhibition but with the (-) enantiomer being more potent (3–10 fold) than the (+) enantiomer. Additionally, both enantiomers of **EU1794-4** appear to compete with **EU1794-27** for access to the binding site, suggesting that enantiomers may interact differently with the same binding site. An alternative explanation for lack of additivity of the effects of the two compounds could be that there are shared residues downstream of the PAM and NAM binding sites that mediate their effects; additional studies are required to evaluate this hypothesis. However, the idea that the enantiomers of the **EU1794** series act at the same site suggests their complex actions on NMDARs expressed in HEK is a mixture of receptors bound to one or the other enantiomer of racemic **EU1794-27** and **EU1794-4**. The available enantiomeric data further support the idea that positive and negative modulators within the **EU1794** series share a single or overlapping binding site.

### EU1794 series links positive and negative allosteric modulators that act at the TMD

An increasing number of NMDAR and AMPAR modulators have been identified with structural determinants of action that reside in transmembrane linker regions and extracellular portions of the transmembrane domain (*Mullasseril et al., 2010*; *Acker et al., 2011*; *Hansen and Traynelis, 2011*;

*Ogden and Traynelis, 2013*; *Yelshanskaya et al., 2016*; *Swanger et al., 2018*; *Wang et al., 2017*). Other cell surface receptor families have bi-directional modulator pockets, including multiple GPCRs as well as the benzodiazepine binding site in GABA-A receptors (*Barnard et al., 1998*; *Rudolph and Knoflach, 2011*; *Wootten et al., 2013*). Among all ionotropic glutamate receptor modulators inter-acting with this region of the receptor, there are positive (CIQ, GNE-9278) and negative (DQP-1105, QNZ-46, CP-465,022, GYKI 52466) modulators with diverse scaffolds. However, the **EU1794** compounds represent the first bidirectional NMDAR modulator series with structural determinants of activity within this region, low micromolar potency and with clear rules to control modulation. The similarity in structure of positive and negative modulators illustrates how subtle differences in the ligand can interconvert functional actions between inhibition and potentiation. It is unclear precisely how the different sized ester alkyl chain substitutions of these analogs bring about opposing actions, further investigation of the structure activity relationship may elucidate these details. The ability to potentiate NMDAR seems to be unique to (-)-**EU1794-27,** which may have a specific interaction with the receptor achieved only by its stereoselective active pose in the binding pocket. Work with this series may lead to an understanding of the mechanistic link between the PAMs and NAMs that inter-act in this portion of the receptor.

## Differential actions on synaptic and extrasynaptic receptors by submaximal EU1794 analogues

The **EU1794** series has a property of agonist-dependence, requiring both glutamate and glycine to be bound before the modulator binding site adopts a high potency orientation for members of the series. Relatedly, use-dependence is a property of open channel blockers (e.g. memantine, MK-801, ketamine, etc.) that inhibit the receptor through interactions within the ion permeation path, and thus rely on pore opening (*Traynelis et al., 2010*). Given that the **EU1794** series are allosteric modu-lators are not voltage-dependent, the mechanism of their agonist-dependence is unclear. Previously, the NAMs QNZ-46, DQP-1105, and NAB-14 have been reported to show varying degrees of gluta-mate- but not glycine-dependence (*Acker et al., 2011*; *Hansen and Traynelis, 2011*; *Swanger et al., 2018*). Moreover, neurosteroid derivatives with NAM activity have also been shown to have glutamate- and glycine-dependence (*Vyklicky et al., 2015*), and the PAM GNE-9278 was reported to be glutamate-dependent (*Wang et al., 2017*). Thus, the **EU1794** series is the first series of positive and negative modulators that has been shown to possess the property of being both glu-tamate- and glycine-dependent. Our working hypothesis is that the **EU1794** series of modulators requires conformational changes in both GluN1 and GluN2 subunits that reflect pre-gating or gating transitions, after which its affinity for its binding site is increased. However, we cannot rule out at this time the possibility that the pore must open to increase modulator binding. Resolving the specific mechanism of the **EU1794** may lead to a more complete understanding of the activation transitions NMDAR.

We believe that **EU1794-4** highlights a novel sub-class of NMDAR modulators that have the capa-bility to selectively act at extrasynaptic NMDARs based on the combinations of its properties. Extra-synaptic NMDAR are hypothesized to respond to glutamate spillover or glial release of glutamate (*Rusakov and Kullmann, 1998*; *Haydon and Carmignoto, 2006*; *Sahlender et al., 2014*). The potential ability to preferentially act at these non-synaptic sites arises from three mechanistic fea-tures of the allosteric mechanism: (1) the submaximal inhibitory effects at saturating concentrations of modulator and agonist, (2) the agonist-dependent and slow association rate, which might limit activity at synaptic receptors, and (3) the ability to enhance NMDAR responses to low agonist con-centrations. A similar property has been described for the GluN2B-selective agent ifenprodil (*Kew et al., 1996*), although the extent to which **EU1794-4** can enhance the response to low con-centrations of agonist is amplified by the large degree of residual current at saturating levels of **EU1794-4** (30–60%) compared to ifenprodil (~10%). There are numerous studies that suggest the importance of extrasynaptic NMDARs in normal biology but study of them requires complex experi-mental paradigms (*Harris and Pettit, 2008*; *Paoletti et al., 2013*; *Papouin and Oliet, 2014*). Alter-natively, the agonist-dependence of **EU1794-4** may alter this capability in instances of repeated stimulation. In either case, **EU1794-4** is a unique compound that may act as a tool that could be used to probe the contribution of distinct types of NMDARs or their activity in circuit function. Fur-ther work is required to fully understand the utility of this modulator as a probe for extrasynaptic NMDARs.

# Materials and methods

## Key resources table

| Reagent type (species) or resource | Designation | Source or reference | Identifiers | Additional information |
|---|---|---|---|---|
| gene (*Rattus norvegicus*) | GluN1-1a | U11418 | | |
| gene (*Rattus norvegicus*) | GluN1-1b | U08263 | | |
| gene (*Rattus norvegicus*) | GluN2A | D13211 | | |
| gene (*Rattus norvegicus*) | GluN2B | U11419 | | |
| gene (*Rattus norvegicus*) | GluN2C | M91563 | | |
| gene (*Rattus norvegicus*) | GluN2D | L31611 | | |
| cell line (*Xenopus laevis*) | stage VI oocytes | Ecocyte Biosciences (Austin, TX); Xenopus 1 (Dexter, MI) | NA; #10004 | |
| cell line (*Homo sapiens*) | HEK293 | ATCC (Manassas, VA) | CRL-1573 | |
| commercial assay or kit | QuikChange kit | Agilent Technologies (Santa Clara, CA) | 200524 | |
| commercial assay or kit | mMessage mMachine SP6 or t7 | Ambion/ThermoFisher Scientific (Waltham, MA) | AM1340 or AM1344 | |
| chemical compound, drug | EU1794-4 | PMID: 26525866 | SMILES: CCOC(=O)C1 = C(NC(=O)CC2SC(=N)NC2 = O)SC2 = C1CCCC2 | |
| chemical compound, drug | EU1794-2 | PMID: 26525866 | SMILES: CCOC(=O)C1 = C(NC(=O)CC2SC(=N)NC2 = O)SC2 = C1CCC(C)C2 | |
| chemical compound, drug | EU1794-5 | this paper | SMILES: CC(C)OC(=O)C1 = C(NC(=O)CC2SC(=N)NC2 = O)SC2 = C1CCCC2 | See supplemental information |
| chemical compound, drug | EU1794-19 | this paper | SMILES: CC(C)OC(=O)C1 = C(NC(=O)CC2SC(=N)NC2 = O)SC2 = C1CCC(C)C2 | See supplemental information |
| chemical compound, drug | EU1794-25 | this paper | SMILES: CC1CCC2 = C(C1)SC(NC(=O)CC1SC(=N)NC1 = O)=C2C(=O)OC1 = CC = CC=C1 | See supplemental information |
| chemical compound, drug | EU1794-27 | this paper | SMILES: CC(C)(C)OC(=O)C1 = C(NC(=O)CC2SC(=N)NC2 = O)SC2 = C1CCCC2 | See supplemental information |
| chemical compound, drug | EU1794-29 | this paper | SMILES: CCCCOC(=O)C1 = C(NC(=O)CC2SC(=N)NC2 = O)SC2 = C1CCC(C)C2 | See supplemental information |
| chemical compound, drug | βCD | Acros Organics or ChemCenter | 2-(hydroxypropyl)-β-cyclodextrin (CAS# 128446-35-5) | |
| chemical compound, drug | APV | Tocris (Minneapolis, MN) | (2)-amino-5-phosphonovaleric acid (CAS# 76326-31-3) | |
| chemical compound, drug | 7-CKA | Tocris (Minneapolis, MN) | 7-chlorokynurenic acid (CAS# 18000-24-3) | |
| chemical compound, drug | MTSEA | TRC Canada (Toronto, ON) | Methanethiosulfonate ethylammonium (CAS# 16599-33-0) | |
| chemical compound, drug | CIQ | Tocris (Minneapolis, MN) | CAS# 486427-17-2 | |
| chemical compound, drug | Ifenprodil | Tocris (Minneapolis, MN) | CAS# 23210-56-2 | |
| chemical compound, drug | PYD-106 | PMID: 25205677 | | |
| chemical compound, drug | TCN-201 | Tocris (Minneapolis, MN) | CAS# 852918-02-6 | |
| chemical compound, drug | GNE-6901 | PMID: 26875626; Genentech (San Francisco, CA) | | generously provided by Jesse Hanson and Genentech |
| chemical compound, drug | GNE-0723 | PMID: 26919761; Genentech (San Francisco, CA) | | generously provided by Jesse Hanson and Genentech |
| chemical compound, drug | DQP-1105 | Tocris (Minneapolis, MN) | CAS# 380560-89-4 | |

*Continued on next page*

*Continued*

| Reagent type (species) or resource | Designation | Source or reference | Identifiers | Additional information |
|---|---|---|---|---|
| chemical compound, drug | QNZ-46 | Tocris (Minneapolis, MN) | CAS# 1237744-13-6 | |
| software, algorithm | Perszyk2018_ sim_Markov_ Modulator Competition | this paper | | Matlab code, see sumplemental information |
| software, algorithm | Axopatch and Clampfit | Molecular Devices (San Jose, CA) | | |
| software, algorithm | Easy Oocyte | Custom written Software (Emory University, Atlanta, GA) | | |
| software, algorithm | Prism | GraphPad (La Jolla, CA) | | |
| software, algorithm | Matlab | Mathworks (Natick, MA) | | |

## Molecular Biology

cDNAs for rat wild type NMDAR subunits GluN1-1a (GenBank U11418, U08261; hereafter GluN1 or GluN1a), GluN1-1b (U08263, hereafter GluN1b), GluN2A (D13211), GluN2B (U11419), GluN2C (M91563), and GluN2D (L31611, modified as described in *Monyer et al., 1994*) were provided by Drs. S. Heinemann (Salk Institute), S. Nakanishi (Kyoto University), and P. Seeburg (University of Heidelberg). Site-directed mutagenesis was conducted using the QuikChange kit (Agilent Technologies, Santa Clara, CA) following the recommended protocol. DNA sequencing was used to verify all mutations. The amino acid numbering system started with the initiating methionine as residue number one. For expression in *Xenopus laevis* oocytes, cDNA constructs were linearized by restriction enzymes, and subsequently used to synthesize in vitro cRNAs following the manufacturer's protocol (mMessage mMachine, Ambion, ThermoFisher Scientific, Waltham, MA).

## Two-electrode voltage-clamp recordings

*Xenopus laevis* stage VI oocytes (Ecocyte Biosciences, Austin, TX) were injected as previously described (*Hansen et al., 2013*) 2–7 days prior to recording with cRNA encoding the NMDAR subunits and stored at 15°C in media containing (in mM) 88 NaCl, 2.4 NaHCO$_3$, 1 KCl, 0.33 Ca(NO$_3$)$_2$, 0.41 CaCl$_2$, 0.82 MgSO$_4$, 5 Tris-HCl; pH was adjusted to 7.4 with NaOH and the solution supplemented with 1 U/mL penicillin, 0.1 mg/mL gentamicin sulfate, and 1 µg/mL streptomycin. Two-electrode voltage-clamp recordings were performed at room temperature (21–23°C) using different internal solutions for voltage and current electrodes (0.3 M and 3.0 M KCl, respectively). All extracellular solutions were made from a solution containing (in mM) 90 NaCl, 1 KCl, 10 HEPES, 0.5 BaCl$_2$, 0.01 EDTA (pH 7.4 with NaOH). Experiments were performed using an eight port Modular Valve Positioner (Hamilton Company, Reno, NV) and controlled by custom software (Easy Oocyte, Emory University, Atlanta, GA)to exchange solutions. Oocyte currents were recorded at a holding potential of −40 mV using a two-electrode voltage-clamp amplifier (OC-725B or OC-725C, Warner Instruments, Hamden, CT). For some recordings, 1 mM 2-(hydroxypropyl)-β-cyclodextrin was added to wash solutions to ensure modulators did not adhere to the tubing, valves, or recording chamber. In MTSEA experiments, fresh MTSEA solutions (0.2 mM) were prepared and utilized in less than 30 min to limit compound degradation.

## Whole cell patch-clamp recordings

HEK293 cells (CRL-1573, ATCC, Manassas, VA) were maintained in a tissue culture room solely reserved for their culturing and transfected using a CaPO$_4$ protocol to express the diheteromeric NMDARs as previously described (*Hansen et al., 2013*). Cell lines were obtained directly from the supplier, were expanded, frozen and kept in liquid nitrogen in a dewar. Mycoplasma testing was not performed. cDNA ratios for transfection were 1:1:1 (GluN2D) or 1:1:5 (GluN2A) for GluN1:GluN2: GFP. Cells were maintained in 5% CO$_2$ at 37°C for 12–36 hr post-transfection. Prior to and during recording, cells were bathed in a solution containing (in mM) 150 NaCl, 10 HEPES, 3 KCl, 0.5 CaCl$_2$, 0.01 EDTA (pH 7.4). Cells were patched with borosilicate glass micropipettes (3–4 MΩ) that were filled with internal recording solution that contained (in mM) 95 CsGluconate, 5 CsCl, 40 HEPES, 8

NaCl, 5 MgCl$_2$, 10 BAPTA, 0.6 EGTA, 2 Na$_2$ATP, and 0.3 NaGTP (pH 7.35). The whole cell recording conformation was achieved, and the cell lifted into the flow of solution from a two barreled theta glass or a triple barrel square glass perfusion system to perform rapid solution exchange experiments. The theta/triple barreled glass was translated using a piezoelectric manipulator to exchange the solution around the cell. Calibration of the perfusion manifold was performed each day to ensure the 10–90% rise time of the solution exchange (switching between 0/100% and 50/50% H$_2$O/external solution) around an open tip was less than 1.5 ms for a theta tube and 4 ms for a transition into an outside lane and 2 ms for a transition between lanes for a triple barreled manifold. For some recordings, 1 mM 2-(hydroxypropyl)-β-cyclodextrin was added to wash solutions to ensure modulators did not adhere to the tubing, valves, or recording chamber.

## Statistical analysis

Concentration-response curves for both positive and negative modulators were fitted by the Hill equation:

$$\frac{I}{I_{[M]=0}} = 1 + Extent\left(\frac{[M]^h}{[M]^h + EC_{50}^h}\right)$$

where $I$ is the measured response of receptor activation, *Extent* is the maximal predicted modulation of the glutamate/glycine response, $[M]$ is the concentration of the modulator being used, $h$ is the Hill slope, and $EC_{50}$ is the half-maximal effective concentration of the modulator. Modulator concentration response data were plotted as a percentage of unmodulated response and displayed fitted curves were obtained by fitting all data simultaneously.

Glutamate and glycine concentration-response curves were fit with the Hill equation as follows:

$$\frac{I}{I_{Max}} = \left(\frac{[A]^h}{[A]^h + EC_{50}^h}\right)$$

where $I$ is the measured response of receptor activation, $[A]$ is the concentration of the agonist, $h$ is the Hill slope, and $EC_{50}$ is the half-maximally effective concentration of the agonist. Agonist concentration-response data, unless otherwise stated, were plotted as a percentage of maximal response and displayed fitted curves were obtained by fitting all data simultaneously with the appropriate Hill equation.

The time course for the onset and offset of positive and negative modulation was fitted using a single exponential function:

$$I_t = A * \left(1 - e^{\frac{-t}{\tau}}\right) + c$$

where $t$ is time, $I_t$ is the difference in current at a given time point as compared to the response at $t$ = 0, $A$ is the amplitude of the exponential, $\tau$ is the time constant for the exponential and $c$ is a constant. Additionally, some of the complex modulator time-courses were fitted with the sum of a single exponential equation and a linear component

$$I_t = A * \left(1 - e^{\frac{-t}{\tau}}\right) + mt + c$$

where $m$ is the slope.

Measurements are given as mean ± SEM unless otherwise indicated. The number of replicate experiments used was chosen to ensure a power level of at least 0.80 when α = 0.05 and when detecting appropriate effect sizes. $EC_{50}$ and $IC_{50}$ values and confidence intervals reported were calculated by averaging the log($EC_{50}$) or log($IC_{50}$) value, determining confidence intervals for mean log($EC_{50}$) or log($IC_{50}$), and converting back to units of molarity. Statistical significance evaluations (α set to 0.05) were performed using a one-way ANOVA with Dunnett's multiple comparison test, a paired t–test, or other appropriate tests as described.

## Modelling receptor function

Matlab (Mathworks, Natick, MA) scripts were written based on previously described approaches to modelling channel function (*Colquhoun and Hawkes, 1977*; *Colquhoun and Hawkes, 1995*). Briefly, a Q matrix was constructed for a model that accounts for macroscopic responses (*Lester and Jahr, 1992*). The glutamate association and dissociation rates were from (*Erreger et al., 2005*), and unidirectional rate constants for gating and modulator binding were chosen to allow the model to be used for modulator competition. The differential equations derived by the Q matrix were solved using an ordinary differential equation solver (ode23s, Matlab) and occupancy of each state was determined at equilibrium given the concentrations of the theoretical ligands.

## Supplemental methods

### Synthetic chemistry experimental details

All reagents were obtained from commercial suppliers and used without further purification unless specified otherwise, and were > 90% pure as determined by the manufacturer. Reaction progress was monitored by thin layer chromatography (TLC) on pre-coated glass plates (silica gel 60 F254, 0.25 mm) or liquid chromatography-mass spectrometry (LC-MS) using an analytical column (ZORBAX Eclipse XDB-C18, 4.6 x 50 mm, 3.5 µm, Agilent). When noted, flash chromatography was performed on a Teledyne (Lincoln, NE) ISCO Combiflash Companion with prepackaged Teledyne Redisep or Silicycle disposable normal phase silica columns. Proton and carbon NMR spectra were recorded on an INOVA-400 (400 MHz), VNMRS 400 (400 MHz), INOVA-600 (600 MHz), or Unity-600 (600 MHz). All chemical shifts are reported in parts per million and referenced to the residual solvent peak. Mass spectra were performed by the Emory University Mass Spectroscopy Center on either a VG 70-S Nier Johnson or JEOL instrument. The IR spectra were acquired with a Nicolet Avatar 370 DTGS. Final compound purity was established either through high-performance liquid chromatography (HPLC) under the conditions listed or by elemental analysis performed by Atlanta Microlab Inc. where C, H, N agreed with proposed structures within ±0.4% of theoretical values unless indicated otherwise.

### Separation of EU1794-4 enantiomers

Semi-preparative separation of **EU1794-4** enantiomers from racemic **EU1794-4** (0.020 g) was accomplished using a ChiralPak OD-RH (21 mm × 250 mm, 5 µm) with the following conditions: 10.4 mL/min flow rate, 10 mL injection volume (2 mg/1 mL), 30% ACN/70% water with 0.1% formic acid over 120 min to afford (−)-**EU1794-4**, $t_R$ 62 min; (+)-**EU1794-4**, $t_R$ 65 min. The enantiomeric excess (ee) was determined using a ChiralPak OD-RH column (4.6 mm × 150 mm, 5 µm) with the following conditions: 1 mL/min flow rate, 5 µL injection volume, 30% ACN/70% water with 0.1% formic acid. (−)-**EU1794-4**: $t_R$ 29 min; 88% ee; $[\alpha]D^{20}$ = −185 (c 1.0, dry CHCl$_3$). (+)-**EU1794-4**: $t_R$ 34 min, 73% ee; $[\alpha]D^{20}$ = +191 (c 1.0, dry CHCl$_3$).

### Separation of EU1794-27 enantiomers

Semi-preparative separation of **EU1794-27** enantiomers from racemic **EU1794-27** (0.020 g) was accomplished using a ChiralPak OD-RH (21 mm × 250 mm, 5 µm) with the following conditions: 10.4 mL/min flow rate, 10 mL injection volume (2 mg/1 mL), 60% ACN/40% water with 0.1% formic acid over 120 min to afford (−)-**EU1794-27**, $t_R$ 55 min; (+)-**EU1794-27**, $t_R$ 59 min. The enantiomeric excess (ee) was determined using a ChiralPak OD-RH column (4.6 mm × 150 mm, 5 µm) with the following conditions: 1 mL/min flow rate, 5 µL injection volume, 40% ACN/60% water with 0.1% formic acid. (−)-**EU1794-27**: $t_R$ 13 min; 88% ee; $[\alpha]D^{20}$ = −172 (c 1.0, dry CHCl$_3$). (+)-**EU1794-27**: $t_R$ 15 min, 46% ee; $[\alpha]D^{20}$ = +178 (c 1.0, dry CHCl$_3$).

### General Procedure A: Preparation of aminothiophene compounds in alcohol solvent

To a solution of aldehyde/ketone (1.0 eq.) in alcohol (0.2-0.4 M, corresponding to the ester moiety for each reaction, see below) was added cyanoacetate (1.1 eq.), elemental sulfur (1.1 eq.), and morpholine (1.4 eq.). The reaction mixture was heated to 50 °C in an oil bath. When TLC or LC-MS indicated complete conversion (12-24 h.), the reaction mixture was concentrated in vacuo. The resulting residue was taken up in EtOAc and washed with water (2x) and brine (2x). The organic layer was

dried over MgSO₄ and filtered. The solvent was removed *in vacuo*. The resulting product was subjected to flash column chromatography.

## General Procedure B: Preparation of aminothiophene compounds in DMF

To a solution of aldehyde/ketone (1.0 eq.) in DMF (1.0 M) was added cyanoacetate (1.1 eq.), elemental sulfur (1.1 eq.), and morpholine (1.4 eq.). The reaction mixture stirred at room temperature for 24 hours and then concentrated *in vacuo*. The resulting residue was taken up in EtOAc and washed with water (3x) and brine (3x). The organic layer was dried over MgSO₄ and filtered. The solvent was removed *in vacuo*. The resulting product was subjected to flash column chromatography.

## General Procedure C: Preparation of (*Z*)-4-oxo-4-(thiophen-2-ylamino)but-2-enoic acid compounds

A solution of the aminothiophene (1.0 eq.) in anhydrous ether was added dropwise to a solution of maleic anhydride (1.0 eq.) in anhydrous ether for a total concentration of 0.1-0.2 M. Reaction was allowed to stir at room temperature or refluxed until complete consumption of starting material was observed by TLC or LC-MS (16 h.-6 days). The resulting precipitate was filtered, washed with ether, and carried on without further purification.

## General Procedure D: Preparation of 2-(2-imino-4-oxothiazolidin-5-yl)-*N*-(thiophen-2-yl)acetamide compounds from (*Z*)-4-oxo-4-(thiophen-2-ylamino)but-2-enoic acid using EDCl

To a solution of carboxylic acid (1 eq.) in dry DMF (0.15 M) was added HOBt (1.1 eq.) and EDCI (1.5 eq.). The reaction mixture was stirred at room temperature for 1 hour or until all materials were in solution. To the solution was added thiourea (1.1 eq.). The reaction mixture was allowed to stir at room temperature for an additional hour and then heated to 60 °C for 24 hours. The reaction mixture was poured into EtOAc and washed with water (2x) and brine (2x). The organic extracts were combined, dried over anhydrous MgSO₄ and concentrated in vacuo. Crude material was dissolved in hot methanol or ethanol and then slowly cooled to room temperature. The resulting precipitate was filtered and washed with cold solvent.

## General Procedure E: Preparation of aryl maleimide compounds

A mixture of carboxylic acid (1.0 eq.), acetic anhydride (3.3-7.0 eq.), and sodium acetate (0.2 eq.) was heated to 75 °C in an oil bath. After consumption of starting material, reaction mixture was cooled to room temperature where it stirred for 12-18 hours. The reaction was diluted with water and extracted with DCM (3x). The organic extracts were dried over MgSO₄, filtered, and concentrated. The resulting product was subjected to flash column chromatography.

## General Procedure F: Preparation of 2-(2-imino-4-oxothiazolidin-5-yl)-*N*-(thiophen-2-yl)acetamide compounds from aryl maleimides

A mixture of maleimide (1.0 eq.) and thiourea (1.0 eq.) in absolute EtOH (0.1 M) stirred at room temperature for 24-48 hours. Reaction mixture was filtered, and solid material was washed with EtOH and DCM.

**Chemical structure 1.** ethyl 2-amino-6-methyl-4,5,6,7-tetrahydrobenzo[*b*]thiophene-3-carboxylate (**11b**).
DOI: https://doi.org/10.7554/eLife.34711.028

Compound **11b** was prepared according to procedure A using 4-methylcyclohexanone (1.6 mL, 13.4 mmol) and ethyl 2-cyanoacetate (1.6 mL, 14.7 mmol) in absolute ethanol (0.2-0.4 M). The crude material was purified by flash column chromatography (ISCO, 120 g silica gel column, 0-10% EtOAc-Hexanes over 25 minutes) to afford the title compound as a white crystalline solid (3.14 g, 98%). ¹H NMR [400 MHz, CDCl₃] δ 5.92 (br s, 2H), 4.26 (q, J = 7.0 Hz, 2H), 2.85-2.90 (m, 1H), 2.53-2.63 (m,

2H), 2.11-2.17 (m, 1H), 1.81-1.88 (m, 2H), 1.29-1.37 (m, 4H), 1.05 (d, J = 6.4 Hz, 3H); $^{13}$C NMR [100 MHz, CDCl$_3$] δ 166.3, 162.0, 132.3, 117.4, 105.8, 59.5, 32.8, 31.3, 29.7, 26.9, 21.7, 14.7; HRMS (ESI) [M+H]$^+$, calc'd for C$_{12}$H$_{18}$NO$_2$S 240.10528, found 240.10521.

**Chemical structure 2.** ethyl 2-amino-4,5,6,7-tetrahydrobenzo[b]thiophene-3-carboxylate (**11c**)
DOI: https://doi.org/10.7554/eLife.34711.029

Compound **11c** was prepared according to procedure A using cyclohexanone (5.3 mL, 50.9 mmol) and ethyl 2-cyanoacetate (6.0 mL, 56.0 mmol) in absolute ethanol (0.2-0.4 M). The crude material was recrystallized from absolute ethanol to afford the title compound as light yellow crystals (10.0 g, 87%). $^1$H NMR [400 MHz, CDCl$_3$] δ 5.96 (br s, 2H), 4.26 (q, J = 7.1 Hz, 2H), 2.73-2.69 (m, 2H), 2.52- 2.48 (m, 2H), 1.81-1.71 (m, 4H), 1.34 (t, J = 7.2 Hz, 3H); $^{13}$C NMR [100 MHz, CDCl$_3$] δ 166.3, 161.9, 132.6, 117.8, 105.9, 59.5, 27.1, 24.7, 23.4, 23.0, 14.7; HRMS (ESI) [M+H]$^+$, calc'd for C$_{11}$H$_{16}$NO$_2$S 226.08963, found 226.08966; IR (solid ATR): 3400, 3294, 2982, 2939, 2895, 1644, 1593, 1488, 1271 cm$^{-1}$.

**Chemical structure 3.** isopropyl 2-amino-6-methyl-4,5,6,7-tetrahydrobenzo[b]thiophene-3-carboxylate (**11f**)
DOI: https://doi.org/10.7554/eLife.34711.030

Compound **11f** was prepared according to procedure A using 4-methylcyclohexanone (2.2 mL, 17.8 mmol) and isopropyl 2-cyanoacetate (2.5 mL, 19.6 mmol) in isopropanol (0.2-0.4 M). The crude material was purified by flash column chromatography (ISCO, 80 g silica gel column, 0-20% EtOAc-Hexanes over 25 minutes) to afford the title compound as an off-white solid (4.42 g, 98%). $^1$H NMR [400 MHz, CDCl$_3$] δ 5.96 (br s, 2H), 5.16 (septet, J = 6.3 Hz, 1H), 2.84-2.91 (m, 1H), 2.52-2.65 (m, 2H), 2.10-2.18 (m, 1H), 1.80-1.90 (m, 2H), 1.25-1.35 (m, 1H), 1.32 (d, J = 6.4 Hz, 3H), 1.31 (d, J = 6.4 Hz, 3H), 1.04 (d, J = 6.7 Hz, 3H); $^{13}$C NMR [100 MHz, CDCl$_3$] δ 165.9, 161.9, 132.3, 117.4, 106.1, 66.8, 32.9, 31.3, 29.6, 27.0, 22.4, 21.7; HRMS (ESI) [M+H]$^+$, calc'd for C$_{13}$H$_{20}$NO$_2$S 254.12093, found 254.12052.

**Chemical structure 4.** isopropyl 2-amino-4,5,6,7-tetrahydrobenzo[b]thiophene-3-carboxylate (**11h**)
DOI: https://doi.org/10.7554/eLife.34711.031

Compound **11h** was prepared according to procedure A using cyclohexanone (2.1 mL, 20.4 mmol) and isopropyl 2-cyanoacetate (2.8 mL, 22.4 mmol) in isopropanol (0.2-0.4 M). The crude material was purified by flash column chromatography (ISCO, 120 g silica gel column, 0-10% EtOAc-Hexanes over 25 minutes) to afford the title compound as a white solid (4.12 g, 84%). $^1$H NMR [400 MHz, CDCl$_3$] δ 6.12 (br s, 2H), 5.16 (septet, J = 6.2 Hz, 1H), 2.69-2.71 (m, 2H), 2.46-2.48 (m, 2H), 1.72-1.77 (m, 4H), 1.31 (d, J = 6.2 Hz, 6H); $^{13}$C NMR [100 MHz, CDCl$_3$] δ 165.7, 162.0, 132.3, 117.4, 105.6, 66.6, 27.0, 24.5, 23.2, 22.9, 22.1; HRMS (ESI) [M+H]$^+$, calc'd for C$_{12}$H$_{18}$NO$_2$S 240.10528, found 240.10549; Elem. Anal. calc'd for C$_{12}$H$_{17}$NO$_2$S, C, 60.22; H, 7.16; N, 5.85, found C, 60.49; H, 7.26; N, 5.79.

**Chemical structure 5.** tert-butyl 2-amino-4,5,6,7-tetrahydrobenzo[b]thiophene-3-carboxylate (**11k**)
DOI: https://doi.org/10.7554/eLife.34711.032

Compound **11k** was prepared according to procedure A using cyclohexanone (1.6 mL, 15.3 mmol) and tert-butyl 2-cyanoacetate (2.4 mL, 16.8 mmol) in absolute ethanol (40.0 mL). The crude material was purified by flash column chromatography (ISCO, 80 g silica gel, 0-20% EtOAc-Hexanes over 25 minutes) to afford the title compound as a yellow oil (2.45 g, 63%). $^1$H NMR [400 MHz, CDCl$_3$] δ 5.83 (br s, 2H), 2.67-2.64 (m, 2H), 2.49-2.46 (m, 2H), 1.76-1.69 (m, 4H), 1.52 (s, 9H); $^{13}$C NMR [100 MHz, CDCl$_3$] δ 165.8, 161.2, 132.7, 117.7, 107.4, 80.2, 28.8, 27.4, 24.8, 23.5, 23.1; HRMS (ESI) [M+H]$^+$, calc'd for C$_{13}$H$_{20}$NO$_2$S 254.12093, found 254.12087; IR (solid ATR): 3434, 3324, 2932, 1732, 1660, 1576, 1484, 1141 cm$^{-1}$.

**Chemical structure 6.** butyl 2-amino-4,5,6,7-tetrahydrobenzo[b]thiophene-3-carboxylate (**11l**)
DOI: https://doi.org/10.7554/eLife.34711.033

Compound **11l** was prepared according to procedure B using cyclohexanone (1.2 mL, 11.6 mmol) and butyl 2-cyanoacetate (1.8 mL, 12.8 mmol). The crude material was purified by flash column chromatography (ISCO, 80 g silica gel, 0-10% EtOAc-Hexanes over 20 minutes) to afford the title compound as a yellow oil (2.35 g, 80%). $^1$H NMR [400 MHz, CDCl$_3$] δ 5.96 (br s, 2H), 4.21 (t, J = 6.6 Hz, 2H), 2.72-2.69 (m, 2H), 2.52-2.49 (m, 2H), 1.82-1.66 (m, 6H), 1.45 (sextet, J = 7.4 Hz, 2H), 0.96 (t, J = 7.3 Hz, 3H); $^{13}$C NMR [100 MHz, CDCl$_3$] δ 166.5, 161.9, 132.6, 117.8, 106.0, 63.5, 31.1, 27.2, 24.7, 23.4, 23.1, 19.6, 114.0; HRMS (ESI) [M+H]$^+$, calc'd for C$_{13}$H$_{20}$NO$_2$S 254.12093, found 254.12061; IR (solid ATR): 3433, 3325, 2932, 2857, 1664, 1576, 1485, 1265 cm$^{-1}$.

**Chemical structure 7.** benzyl 2-amino-4,5,6,7-tetrahydrobenzo[b]thiophene-3-carboxylate (**11m**)
DOI: https://doi.org/10.7554/eLife.34711.034

Compound **11m** was prepared according to procedure B using cyclohexanone (0.7 mL, 6.5 mmol) and benzyl 2-cyanoacetate (1.1 mL, 7.1 mmol). The crude material was purified by flash column chromatography (ISCO, 40 g silica gel, 0-20% EtOAc-Hexanes over 15 minutes) to afford the title compound as a yellow tinted oil (1.38 g, 74%). $^1$H NMR [400 MHz, CDCl$_3$] δ 7.43-7.33 (m, 5H), 6.00 (br s, 2H), 5.29 (s, 2H), 2.74-2.71 (m, 2H), 2.52-2.49 (m, 2H), 1.81-1.68 (m, 4H); $^{13}$C NMR [100 MHz, CDCl$_3$] δ 166.0, 162.4, 136.9, 132.5, 128.7, 128.1, 128.1, 117.8, 105.5, 65.4, 27.2, 24.7, 23.4, 23.0; HRMS (ESI) [M+H]$^+$, calc'd for C$_{16}$H$_{18}$NO$_2$S 288.10528, found 288.10519; IR (solid ATR): 3442, 3325, 2980, 2934, 1659, 1575, 1483, 1261 cm$^{-1}$.

**Chemical structure 8.** (Z)-4-((3-(ethoxycarbonyl)-6-methyl-4,5,6,7-tetrahydrobenzo[b]thiophen-2-yl)amino)-4-

oxobut-2-enoic acid (**13b**)
DOI: https://doi.org/10.7554/eLife.34711.035

Compound **13b** was prepared according to procedure C using compound **11b** (2.6 g, 10.7 mmol). The crude yellow solid (3.54 g, 98%) was carried on without further purification. $^1$H NMR [400 MHz, CDCl$_3$] δ 12.22 (s, 1H), 6.51 (d, J = 12.8 Hz, 1H), 6.45 (d, J = 12.8 Hz, 1H), 4.38 (q, J = 7.0 Hz, 2H), 2.95-2.99 (m, 1H), 2.78 (dd, J = 4.9, 15.9 Hz, 1H), 2.66-2.75 (m, 1H), 2.29-2.35 (m, 1H), 1.89-1.93 (m, 2H), 1.42 (t, J = 7.3 Hz, 4H), 1.09 (d, J = 6.7 Hz, 3H); $^{13}$C NMR [100 MHz, CDCl$_3$] δ 166.9, 164.3, 162.1, 144.4, 137.8, 132.1, 130.8, 130.0, 115.0, 61.6, 32.8, 31.0, 29.3, 26.3, 21.5, 14.4; HRMS (ESI) [M+H]$^+$, calc'd for C$_{16}$H$_{20}$NO$_5$S, 338.10567, found 338.10550.

**Chemical structure 9.** (*Z*)-4-((3-(ethoxycarbonyl)-4,5,6,7-tetrahydrobenzo[*b*]thiophen-2-yl)amino)-4-oxobut-2-enoic acid (**13c**)
DOI: https://doi.org/10.7554/eLife.34711.036

Compound **13c** was prepared according to procedure C using compound **11c** (3.0 g, 13.3 mmol). The crude yellow solid (4.02 g, 93%) was carried on without further purification. $^1$H NMR [400 MHz, CDCl$_3$] δ 12.18 (br s, 1H), 6.49 (d, J = 12.8 Hz, 1H), 6.45 (d, J = 12.8 Hz, 1H,), 4.36 (q, J = 7.1 Hz, 2H), 2.78 (app. t, J = 5.2 Hz, 2H), 2.69 (app. t, J = 4.9 Hz, 2H), 1.78-1.83 (m, 4H), 1.40 (t, J = 7.3 Hz, 3H); $^{13}$C NMR [100 MHz, CDCl$_3$] δ 166.8, 164.3, 162.0, 144.2, 137.7, 132.3, 131.0, 130.0, 115.1, 61.5, 26.4, 24.7, 22.9, 22.7, 14.4; HRMS (ESI) [M+H]$^+$, calc'd for C$_{15}$H$_{18}$NO$_5$S 324.09002, found 324.08965.

**Chemical structure 10.** (*Z*)-4-((3-(isopropoxycarbonyl)-6-methyl-4,5,6,7-tetrahydrobenzo[*b*]thiophen-2-yl)amino)-4-oxobut-2-enoic acid (**13f**)
DOI: https://doi.org/10.7554/eLife.34711.037

Compound **13f** was prepared according to procedure C using compound **11f** (2.0 g, 7.9 mmol). The crude yellow solid (2.69 g, 97%) was carried on without further purification. $^1$H NMR [400 MHz, CDCl$_3$] δ 12.25 (br s, 1H), 6.49 (d, J = 12.8 Hz, 1H), 6.45 (d, J = 12.8 Hz, 1H), 5.22 (septet, J = 6.2 Hz, 1H), 2.93-2.97 (m, 1H), 2.64-2.79 (m, 2H), 2.26-2.33 (m, 1H), 1.84-1.92 (m, 2H), 1.29-1.43 (m, 7H), 1.08 (d, J = 6.1 Hz, 3H); $^{13}$C NMR [100 MHz, CDCl$_3$] δ 166.4, 164.3, 162.0, 144.2, 137.8, 132.0, 130.7, 130.1, 115.3, 69.5, 32.7, 31.0, 29.3, 26.3, 22.1, 21.5; HRMS (ESI) [M–H]$^-$, calc'd for C$_{17}$H$_{22}$NO$_5$S 350.10677, found 350.10666.

**Chemical structure 11.** (*Z*)-4-((3-(isopropoxycarbonyl)-4,5,6,7-tetrahydrobenzo[*b*]thiophen-2-yl)amino)-4-oxobut-2-enoic acid (**13h**)
DOI: https://doi.org/10.7554/eLife.34711.038

Compound **13h** was prepared according to procedure C using compound **11h** (1.5 g, 6.3 mmol). The crude yellow solid (2.01 g, 95%) was carried on without further purification. $^1$H NMR [600 MHz,

CDCl$_3$] δ 12.27 (br s, 1H), 6.50 (d, J = 12.8 Hz, 1H), 6.46 (d, J = 12.8 Hz, 1H), 5.23 (septet, J = 6.3 Hz, 1H), 2.79-2.80 (m, 2H), 2.69-2.70 (m, 2H), 1.81-1.82 (m, 4H), 1.39 (d, J = 6.0 Hz, 6H); $^{13}$C NMR [150 MHz, CDCl$_3$] δ 166.5, 164.3, 162.1, 144.1, 137.8, 132.4, 131.1, 130.1, 115.5, 69.5, 26.6, 24.7, 22.9, 22.8, 22.2; HRMS (ESI) [M+H]$^+$, calc'd for C$_{16}$H$_{20}$NO$_5$S 338.10567, found 338.10578.

**Chemical structure 12.** (Z)-4-((3-(tert-butoxycarbonyl)-4,5,6,7-tetrahydrobenzo[b]thiophen-2-yl)amino)-4-oxobut-2-enoic acid (13k)
DOI: https://doi.org/10.7554/eLife.34711.039

Compound **13k** was prepared according to procedure C using compound **11k** (2.4 g, 9.6 mmol). The crude yellow solid (2.88 g, 86%) was carried on without further purification. HRMS (ESI) [M+H]$^+$, calc'd for C$_{17}$H$_{22}$NO$_5$S 352.12132, found 352.12107.

**Chemical structure 13.** ethyl 2-(2,5-dioxo-2,5-dihydro-1H-pyrrol-1-yl)-6-methyl-4,5,6,7- tetrahydrobenzo[b]thiophene-3-carboxylate (18b)
DOI: https://doi.org/10.7554/eLife.34711.040

Compound **18b** was prepared according to procedure E using compound **13b** (1.0 g, 3.0 mmol). The crude material was purified by flash column chromatography (ISCO, 24 g silica gel, 0-18% EtOAc-Hexanes over 15 minutes) to afford the title compound as a yellow oil that solidified upon standing (0.769 g, 81%). HRMS (APCI) [M+H]$^+$, calc'd for C$^{16}$H$_{18}$NO$_4$S 320.09511, found 320.09520.

**Chemical structure 14.** isopropyl 2-(2,5-dioxo-2,5-dihydro-1H-pyrrol-1-yl)-4,5,6,7-tetrahydrobenzo[b]thiophene-3-carboxylate (18h)
DOI: https://doi.org/10.7554/eLife.34711.041

Compound **18h** was prepared according to procedure E using compound **13h** (0.8 g, 2.4 mmol). The crude material was purified by flash column chromatography (ISCO, 24 g silica gel, 0-25% EtOAc-Hexanes over 20 minutes) to afford the title compound as a golden brown oil (0.712 g, 94%). $^1$H NMR [400 MHz, CDCl$_3$] δ 6.91 (s, 2H), 5.09 (septet, J = 5.3 Hz, 1H), 2.84 (app. t, J = 5.8 Hz, 2H), 2.75 (app. t, J = 5.5 Hz, 2H), 1.80-1.86 (m, 4H), 1.20 (d, J = 6.1Hz, 6H); HRMS (APCI) [M+H]$^+$, calc'd for C$_{16}$H$_{18}$NO$_4$S 320.09511, found 320.09527.

**Chemical structure 15.** tert-butyl 2-(2,5-dioxo-2,5-dihydro-1H-pyrrol-1-yl)-4,5,6,7-tetrahydrobenzo[b]thiophene-3-carboxylate (18k)
DOI: https://doi.org/10.7554/eLife.34711.042

Compound **18k** was prepared according to procedure E using compound **13k** (2.0 g, 5.7 mmol). The crude material was purified by flash column chromatography (ISCO, 40 g silica gel, 0-18%

EtOAc-Hexanes over 15 minutes) to afford the title compound as a light yellow solid (1.44 g, 76%). [1]H NMR [400 MHz, CDCl$_3$] δ 6.90 (s, 2H), 2.83-2.79 (m, 2H), 2.75-2.72 (m, 2H), 1.86-1.76 (m, 4H), 1.44 (s, 9H); [13]C NMR [100 MHz, CDCl$_3$] δ 169.1 (2C), 161.4, 136.6, 135.7, 135.1 (2C), 131.6, 130.4, 81.6, 28.4 (3C), 26.6, 25.4, 22.9, 22.6; HRMS (APCI) [M–O$t$-Bu]$^+$, calc'd for C$_{13}$H$_{10}$NO$_3$S 260.03759, found 260.03737; IR (solid ATR): 2918, 1709, 1565, 1410, 1393, 1367, 1136, 691 cm$^{-1}$.

**Chemical structure 16.** butyl 2-(2,5-dioxo-2,5-dihydro-1$H$-pyrrol-1-yl)-4,5,6,7-tetrahydrobenzo[$b$]thiophene-3-carboxylate (**18l**)
DOI: https://doi.org/10.7554/eLife.34711.043

Compound **18l** was prepared according to procedure E using compound **13l** (1.2 g, 3.4 mmol). The crude material was purified by flash column chromatography (ISCO, 40 g silica gel, 0-18% EtOAc-Hexanes over 15 minutes) to afford the title compound as a light yellow solid (1.04 g, 91%). [1]H NMR [400 MHz, CDCl$_3$] δ 6.91 (s, 2H), 4.13 (t, $J$ = 6.7 Hz, 2H), 2.84 (app. t, $J$ = 5.8 Hz, 2H), 2.75 (app. t, $J$ = 5.8 Hz, 2H), 1.87-1.78 (m, 4H), 1.57 (p, $J$ = 7.2 Hz, 2H), 1.34 (sextet, $J$ = 7.5 Hz, 2H), 0.91 (t, $J$ = 7.5 Hz, 3H); [13]C NMR [100 MHz, CDCl$_3$] δ 169.2 (2C), 162.21, 136.8, 135.9, 135.1 (2C), 133.0, 128.7, 64.7, 30.8, 26.6, 25.4, 22.8, 22.6, 19.3, 13.9; 62 HRMS (APCI) [M+H]$^+$, calc'd for C$_{17}$H$_{20}$NO$_4$S 334.11076, found 334.11128; IR (solid ATR): 3097, 2944, 1710, 1414, 1364, 1175, 836, 693 cm$^{-1}$.

**Chemical structure 17.** benzyl 2-(2,5-dioxo-2,5-dihydro-1$H$-pyrrol-1-yl)-4,5,6,7-tetrahydrobenzo[$b$]thiophene-3-carboxylate (**18m**)
DOI: https://doi.org/10.7554/eLife.34711.044

Compound **18m** was prepared according to procedure E using compound **13m** (0.4 g, 1.0 mmol). The crude material was purified by flash column chromatography (ISCO, 24 g silica gel, 0-25% EtOAc-Hexanes over 15 minutes) to afford the title compound as a light yellow solid (0.299 g, 78%). [1]H NMR [400 MHz, CDCl$_3$] δ 7.40-7.21 (m, 5H), 6.44 (s, 2H), 5.07 (s, 2H), 2.91-2.78 (m, 2H), 2.75-2.64 (m, 2H), 1.88-1.71 (m, 4H); [13]C NMR [100 MHz, CDCl$_3$] δ 169.2 (2C), 161.6, 137.0, 136.3, 135.4, 134.2 (2C), 133.3, 129.4 (2C), 128.8 (2C), 128.7, 128.3, 67.0, 26.4, 25.4, 22.8, 22.5; HRMS (APCI) [M+H]$^+$, calc'd for C$_{20}$H$_{18}$NO$_4$S 368.09511, found 368.09512; IR (solid ATR): 2942, 2850, 1763, 1672, 1638, 1537, 1176, 1137, 691 cm$^{-1}$.

**Chemical structure 18.** ($Z$)-4-((3-((benzyloxy)carbonyl)-4,5,6,7-tetrahydrobenzo[$b$]thiophen-2-yl)amino)-4-oxobut-2-enoic acid (**13m**)
DOI: https://doi.org/10.7554/eLife.34711.045

Compound **13m** was prepared according to procedure C using compound **11m** (0.6 g, 2.1 mmol). The crude yellow solid (0.573 g, 70%) was carried on without further purification. HRMS (ESI) [M+H]$^+$, calc'd for C$_{20}$H$_{20}$NO$_5$S 386.10567, found 386.10587.

**Chemical structure 19.** ethyl 2-(2-(2-imino-4-oxothiazolidin-5-yl)acetamido)-6-methyl-4,5,6,7-tetrahydrobenzo[*b*] thiophene-3-carboxylate (**EU1794-2**)
DOI: https://doi.org/10.7554/eLife.34711.046

Compound **EU1794-2** was prepared according to procedure D using compound **13b** (0.3 g, 0.7 mmol). The crude material was precipitated from hot ethanol to afford the title compound (inseparable mixture of diastereomers) as a light yellow solid (0.0779 g, 27%). Compound **EU1794-2** was also prepared according to procedure F using compound **18b** (0.4 g, 1.3 mmol). The precipitate was rinsed with DCM to afford the title compound (inseparable mixture of diastereomers) as an off-white solid (0.406 g, 82%). $^1$H NMR [400 MHz, DMSO-$d_6$] δ 10.98 (s, 1H), 9.03 (s, 1H), 8.80 (s, 1H), 4.42 (dd, $J$ = 3.7, 10.4 Hz, 1H), 4.27 (q, $J$ = 6.9 Hz, 2H), 2.99 (dd, $J$ = 10.4, 15.3 Hz, 1H), 2.83-2.88 (m, 1H), 2.59-2.70 (m, 2H), 2.16-2.22 (m, 1H), 1.80 (m, 2H), 1.30 (t, $J$ = 7.0 Hz, 4H), 1.01 (d, $J$ = 6.7 Hz, 4H); $^{13}$C NMR [100 MHz, DMSO-$d_6$] ($^{13}$C resonance obstructed by solvent peak) δ 188.5, 182.0, 167.8, 164.7, 145.4, 130.2, 125.9, 111.7, 60.4, 51.4, 31.7, 30.5, 28.7, 25.6, 21.2, 14.1; HRMS (ESI) [M+H]$^+$, calc'd for $C_{13}H_{16}N_3O_4S_2$, 342.05768, found 342.05759; IR (solid ATR): 3224, 2927, 1673, 1562, 1534, 1496, 1460, 1435, 1376, 1284, 1257, 1231, 1166, 1142, 1081, 1027, 960, 843, 775, 741, 698, 636 cm$^{-1}$; m.p. = 190-204 °C (dec.); Elem. Anal. calc'd for $C_{13}H_{15}N_3O_4S_2$, C, 45.73; H, 4.43; N, 12.31, found C, 45.88; H, 4.46; N, 11.93.

**Chemical structure 20.** ethyl 2-(2-(2-imino-4-oxothiazolidin-5-yl)acetamido)-4,5,6,7-tetrahydrobenzo[*b*]thiophene-3-carboxylate (**EU1794-4**)
DOI: https://doi.org/10.7554/eLife.34711.047

Compound **EU1794-4** was prepared according to procedure D using compound **13c** (0.4 g, 1.2 mmol). The crude material was precipitated from hot ethanol to afford the title compound as a light yellow solid (0.281 g, 60%). $^1$H NMR [400 MHz, DMSO-$d_6$] δ 11.00 (s, 1H), 9.06 (s, 1H), 8.82 (s, 1H), 4.43 (dd, $J$ = 3.9, 10.2 Hz, 1H), 4.28 (q, $J$ = 7.0 Hz, 2H), 3.36-3.42 (m, 1H), 3.01 (dd, $J$ = 10.2, 16.8, 1H), 2.69 (s, 2H), 2.59 (s, 2H), 1.72 (s, 4H), 1.31 (t, $J$ = 7.0 Hz, 3H); $^{13}$C NMR [100 MHz, DMSO-$d_6$] ($^{13}$C resonance obstructed by solvent peak) δ 188.5, 182.0, 167.8, 164.7, 145.3, 130.5, 126.3, 111.8, 60.3, 51.4, 25.8, 23.7, 22.5, 22.3, 14.1; HRMS (ESI) [M+H]$^+$, calc'd for $C_{16}H_{20}N_3O_4S_2$, 382.08952, found 382.08902; IR (solid ATR): 3661, 3269, 3195, 2933, 1655, 1563, 1527, 1439, 1408, 1366, 1326, 1231, 1134, 1032, 996, 969, 784, 703, 629 cm$^{-1}$; m.p. = 163 °C (dec.); Elem. Anal. calc'd for $C_{16}H_{19}N_3O_4S_2$, C, 50.38; H, 5.02; N, 11.02, found C, 49.31; H, 5.00; N, 10.73 (Value did not fall within ± 0.4).

**Chemical structure 21.** isopropyl 2-(2-(2-imino-4-oxothiazolidin-5-yl)acetamido)-6-methyl-4,5,6,7-tetrahydrobenzo [*b*]thiophene-3-carboxylate (**EU1794-19**)
DOI: https://doi.org/10.7554/eLife.34711.048

Compound **EU1794-19** was prepared according to procedure D using compound **13f** (0.7 g, 2.0 mmol). The crude material was precipitated from hot ethanol to afford the title compound as a white solid (0.134 g, 16%). [1]H NMR [400 MHz, DMSO-$d_6$] δ 11.00 (s, 1H), 9.04 (s, 1H), 8.81 (s, 1H), 4.43 (dd, J = 3.8, 10.2 Hz, 1H), 4.29 (q, J = 7.1 Hz, 2H), 3.39 (dd, J = 3.8, 16.9 Hz, 1H), 3.00 (dd, J = 10.4, 16.8 Hz, 1H), 2.71 (app. t, J = 5.9 Hz, 2H), 2.38 (app. s, 2H), 1.48 (app. t, J = 6.3 Hz, 2H), 1.32 (t, J = 7.0 Hz, 3H), 0.95 (s, 6H).; [13]C NMR [100 MHz, DMSO-$d_6$] δ 188.5, 181.9, 167.7, 164.7, 145.5, 129.1, 125.6, 111.5, 60.4, 51.4, 39.2, 37.1, 35.0, 29.7, 27.4, 27.3, 23.5, 14.0; HRMS (ESI) [M+H]+, calc'd for $C_{16}H_{24}N_3O_4S_2$ 410.12028, found 410.12066; IR (solid ATR): 3205, 2923, 1655, 1561, 1531, 1492, 1386, 1293, 1233, 1148, 1106, 1074, 1011, 979, 960, 912, 835, 783, 761, 736, 698, 645 cm$^{-1}$; m.p. = 229 °C (dec.); HPLC purity: >95% $t_R$ = 1.300 min. (85% MeOH/Water with 0.1% formic acid, isocratic, 1.0 mL/min.); $t_R$ = 0.995 min. (75% MeCN/Water with 0.1% formic acid, isocratic, 1.0 mL/min.).

**Chemical structure 22.** isopropyl 2-(2-(2-imino-4-oxothiazolidin-5-yl)acetamido)-4,5,6,7-tetrahydrobenzo[*b*]thiophene-3-carboxylate (**EU1794-5**)
DOI: https://doi.org/10.7554/eLife.34711.049

To a solution of **13h** (0.2 g, 0.6 mmol) dissolved in dry DMF (5.0 mL) was added HATU (0.2 g, 0.6 mmol). The reaction mixture was allowed to stir at room temperature for 10 minutes. To the stirring mixture was added thiourea (0.1 g, 1.5 mmol) and DIPEA (0.5 mL, 3.0 mmol). The reaction was allowed to stir an additional 30 minutes before heating to 60 °C for 72 hours. The solvent was removed *in vacuo*. The crude material was purified by flash column chromatography (ISCO, 12 g silica gel column, 3-11% MeOH-DCM over 15 minutes). The resulting fractions were found to contain impurities. Material was re-purified by flash column chromatography (12 g silica gel column, 0-8% MeOH-DCM over 15 minutes) to afford the title compound as a light brown solid (0.0551 g, 23%). Compound **EU1794-5** was also prepared according to procedure F using **18h** (0.2 g, 0.7 mmol). The precipitate was rinsed with DCM to afford the title compound as a white solid (0.119 g, 42%). [1]H NMR [400 MHz, DMSO-$d_6$] δ 11.04 (s, 1H), 9.04 (s, 1H), 8.81 (s, 1H), 5.10 (septet, J = 6.3 Hz, 1H), 4.42 (dd, J = 3.9, 10.2 Hz, 1H), 3.40 (d, J = 3.5 Hz, 1H), 3.00 (dd, J = 10.2 16.8, 1H), 2.69 (br s, 2H), 2.58 (br s, 2H), 1.71 (br s, 4H), 1.31 (d, J = 6.3 Hz, 6H); [13]C NMR [100 MHz, DMSO-$d_6$] ([13]C resonance obstructed by solvent peak) δ 188.5, 181.9, 167.7, 164.4, 145.4, 130.4, 126.3, 112.0, 68.0, 51.4, 25.9, 23.7, 22.5, 22.3, 21.7 (2C); HRMS (ESI) [M+H]+, calc'd for $C_{17}H_{22}N_3O_4S_2$ 396.10463, found 396.10409; m.p. = 152-154 °C; HPLC purity: >95% $t_R$ = 2.082 min. (75-95% MeOH/Water with 0.1% formic acid, gradient over 3 minutes, 1.0 mL/min.); $t_R$ = 0.698 min. (85% MeCN/Water with 0.1% formic acid, isocratic, 1.0 mL/min.).

**Chemical structure 23.** tert-butyl 2-(2-(2-imino-4-oxothiazolidin-5-yl)acetamido)-4,5,6,7-tetrahydrobenzo[*b*]thiophene-3-carboxylate (**EU1794-27**)
DOI: https://doi.org/10.7554/eLife.34711.050

Compound **EU1794-27** was prepared according to procedure F using compound **18k** (0.5 g, 1.5 mmol). The precipitate was rinsed with EtOH and EtOAc to afford the title compound as a light yellow solid (0.403 g, 66%). [1]H NMR [400 MHz, DMSO-$d_6$] δ 11.03 (s, 1H), 9.03 (s, 1H), 8.80 (s, 1H), 4.43 (dd, J = 4.0, 10.2 Hz, 1H), 3.38 (dd, J = 3.7, 16.8 Hz, 1H), 3.00 (dd, J = 10.4, 16.8 Hz, 1H), 2.68 (app. br s, 2H), 2.58 (app. br s, 2H), 1.70 (app. br s, 4H), 1.54 (s, 9H); [13]C NMR [100 MHz, DMSO-$d_6$] ([13]C resonance obstructed by solvent peak) δ 188.5, 182.0, 167.7, 164.3, 145.2, 130.5, 126.1, 112.8, 81.5, 51.4, 28.0 (3C), 26.1, 23.8, 22.5, 22.4; HRMS (ESI) [M+H]+, calc'd for $C_{18}H_{24}N_3O_4S_2$ 410.12027,

found 410.12011; IR (solid ATR): 3372, 2919, 1705, 1565, 1524, 1367, 1240, 1138 cm$^{-1}$; HPLC purity: >95% $t_R$ = 0.765 min. (95% MeOH/Water with 0.1% formic acid, isocratic, 1.0 mL/min.).

**Chemical structure 24.** butyl 2-(2-(2-imino-4-oxothiazolidin-5-yl)acetamido)-4,5,6,7-tetrahydrobenzo[*b*]thiophene-3-carboxylate (EU1794-29)

DOI: https://doi.org/10.7554/eLife.34711.051

Compound **EU1794-29** was prepared according to procedure F using compound **18l** (0.5 g, 1.5 mmol). The precipitate was rinsed with EtOH to afford the title compound as a light yellow solid (0.325 g, 53%). $^1$H NMR [400 MHz, DMSO-$d_6$] δ 11.38 (s, 1H), 10.05 (br s, 1H), 7.53 (s, 1H), 4.55 (dd, J = 3.2, 11.1 Hz, 1H), 4.24 (t, J = 6.6 Hz, 2H), 3.54 (dd, J = 3.1, 16.8 Hz, 1H), 2.96 (dd, J = 11.3, 16.8 Hz, 1H), 2.72 (app. br s, 2H), 2.57 (app. br s, 2H), 1.76 (app. br s, 4H), 1.70 (p, J = 7.2 Hz, 2H), 1.44 (sextet, J = 7.4 Hz, 2H), 0.96 (t, J = 7.3 Hz, 3H); $^{13}$C NMR [100 MHz, DMSO-$d_6$] δ 189.1, 184.3, 167.0, 166.7, 146.9, 131.0, 127.2, 112.2, 64.7, 52.0, 40.2, 30.9, 26.5, 24.5, 23.1, 23.0, 19.6, 13.9; HRMS (ESI) [M+H]$^+$, calc'd for $C_{18}H_{24}N_3O_4S_2$ 410.12027, found 410.11979; IR (solid ATR): 3307, 2939, 1650, 1567, 1521, 1414, 1245, 1115 cm$^{-1}$; HPLC purity: >95% $t_R$ = 1.479 min. (85% MeOH/Water with 0.1% formic acid, isocratic, 1.0 mL/min.).

**Chemical structure 25.** benzyl 2-(2-(2-imino-4-oxothiazolidin-5-yl)acetamido)-4,5,6,7-tetrahydrobenzo[*b*]thiophene-3-carboxylate (EU1794-25)

DOI: https://doi.org/10.7554/eLife.34711.052

Compound **EU1794-25** was prepared according to procedure F using compound **18m** (0.3 g, 0.7 mmol). The precipitate was rinsed with DCM to afford the title compound as an off-white solid (0.124 g, 41%). $^1$H NMR [400 MHz, DMSO-$d_6$] δ 10.98 (s, 1H), 9.04 (s, 1H), 8.80 (s, 1H), 7.49-7.29 (m, 5H), 5.33 (s, 2H), 4.42 (dd, J = 3.8, 10.2 Hz, 1H), 3.37 (dd, J = 4.0, 16.8 Hz, 1H), 2.96 (dd, 1H, J = 10.2, 16.9 Hz, 1H), 2.72-2.64 (m, 2H), 2.62-2.54 (m, 2H), 1.77-1.61 (m, 4H); $^{13}$C NMR [100 MHz, DMSO-$d_6$] (13C resonance obstructed by solvent peak) δ 188.5, 181.97, 167.9, 164.4, 145.7, 136.1, 130.5, 128.6 (2C), 128.2, 128.1 (2C), 126.4, 111.6, 65.9, 51.4, 25.9, 23.7, 22.5, 22.3; HRMS (ESI) [M+H]$^+$, calc'd for $C_{21}H_{22}N_3O_4S_2$ 444.10463, found 444.10497; IR (solid ATR): 2936, 1649, 1564, 1517, 1237, 1191, 1020 cm$^{-1}$; HPLC purity: >95% $t_R$ = 2.470 min. (75-95% MeOH/Water with 0.1% formic acid, gradient over 3 minutes, 1.0 mL/min.).

## Acknowledgements

We thank Phuong Le, Jing Zhang, and Anel Tankovic for excellent technical assistance. GNE-6901 and GNE-0723 were generously provided by Jesse Hanson and Genentech. This work was supported by the NINDS (NS036654, NS065371, SFT) and NIGMS (T32-GM008602). We also thank the Emory Chemical Biology Discovery Center for their assistance.

## Additional information

### Competing interests

Brooke M Katzman: BMK is an inventor of Emory-owned IP. Matthew P Epplin: MPE is an inventor of Emory-owned IP. David Menaldino: DM is an inventor of Emory-owned IP. Dennis C Liotta: DCL is a member of the Board of Directors at NeurOp Inc. and is an inventor of Emory-owned IP. Stephen F

Traynelis: SFT is a co-founder of NeurOp Inc., a paid consultant for Janssen, a PI on a research grant from Janssen to Emory University, a member of the Scientific Advisory Board for Sage Therapeutics, and is an inventor of Emory-owned IP. The other authors declare that no competing interests exist.

## Funding

| Funder | Grant reference number | Author |
| --- | --- | --- |
| National Institute of Neurological Disorders and Stroke | NS036654,NS065371 | Stephen F Traynelis |
| National Institute of General Medical Sciences | GM008602 | Riley Perszyk |

The funders had no role in study design, data collection and interpretation, or the decision to submit the work for publication.

## Author contributions

Riley Perszyk, Conceptualization, Data curation, Software, Formal analysis, Validation, Investigation, Visualization, Methodology, Writing—original draft, Writing—review and editing; Brooke M Katzman, Stephen F Traynelis, Conceptualization, Resources, Supervision, Funding acquisition, Writing—original draft, Writing—review and editing; Hirofumi Kusumoto, Conceptualization, Resources, Data curation, Software, Formal analysis, Validation, Investigation, Visualization, Methodology, Writing—original draft, Writing—review and editing; Steven A Kell, Conceptualization, Resources, Writing—original draft, Writing—review and editing; Matthew P Epplin, Rhonda L Moore, David Menaldino, Pieter Burger, Conceptualization, Resources, Writing—review and editing; Yesim A Tahirovic, Resources, Investigation, Writing—review and editing; Dennis C Liotta, Conceptualization, Resources, Supervision, Funding acquisition, Writing—review and editing

## Author ORCIDs

Riley Perszyk (iD) https://orcid.org/0000-0001-8283-7709
Stephen F Traynelis (iD) https://orcid.org/0000-0002-3750-9615

## Decision letter and Author response

Decision letter https://doi.org/10.7554/eLife.34711.057
Author response https://doi.org/10.7554/eLife.34711.058

# Additional files

## Supplementary files

• Supplementary file 1. Saturating and sub-saturating actions of EU1794-2.
DOI: https://doi.org/10.7554/eLife.34711.053

• Supplementary file 2. Enantiomeric actions of EU1794-4.
DOI: https://doi.org/10.7554/eLife.34711.054

• Transparent reporting form
DOI: https://doi.org/10.7554/eLife.34711.055

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
