## [Decision Letter]

Thank you for submitting your article "An NMDAR positive and negative allosteric modulator series share a binding site and are interconverted by methyl groups" for consideration by *eLife*. Your article has been reviewed by three peer reviewers, and the evaluation has been overseen by Kenton Swartz as the Reviewing Editor and Richard Aldrich as the Senior Editor. The following individuals involved in review of your submission have agreed to reveal their identity: Alexander Sobolevsky (Reviewer #1).

The reviewers have discussed the reviews with one another and the Reviewing Editor has drafted this decision to help you prepare a revised submission. As you can see below, all three reviewers are enthusiastic about your manuscript and recommend publication in *eLife* once you have had the chance to address their relatively minor points; most of these can be addressed with careful editing of the manuscript. The one request for additional data concerns the open probability of a mutant that influences the activity of your allosteric ligands, which would be nice to include, but we leave this to your discretion and would accept inclusion of a discussion of this issue.

Reviewer #1:

NMDA receptors (NMDARs) are prominent elements of excitatory neurotransmission, important for learning and memory and implicated in a plethora of neurological disorders. Despite the enormous role in pathophysiology, attempts to design drugs targeting NMDARs had limited success, mainly due to a variety of accompanying side effects. Perszyk et al., describe a novel class of positive and negative allosteric modulators (PAMs and NAMs) and decipher the mechanism of their action, which suggests possibility of using this class of small molecules for future medicinal purposes. The uniqueness of the mechanism is that the small molecules of the same EU1794 scaffold, depending on decoration of the ester linkage to the scaffold by aliphatic substitutions and the NR2 subunit subtype, through binding to the same site at the ion channel extracellular collar can produce inhibition or potentiation of NMDAR-mediated currents that require receptor activation but can further be tuned by concentrations of agonist glutamate and co-agonist glycine. The new PAMs and NAMs therefore represent the first series of high potency bidirectional NMDAR modulators that have a potential to preserve NMDAR-mediated synaptic transmission, but to modulate extrasynaptic NMDARs in disease-specific ways. The study includes an impressive amount of data and the same time plenty of very elegant experiments. The experimental results are of high quality and interpretations are just.

Reviewer #2:

NMDA receptors play a central role in brain physiology and function as well as in numerous neurological disorders such as epilepsy, Alzheimer's disease, and schizophrenia to name just a small sample. The development of pharmacological tools to target NMDA receptors in the clinic as well as to understand their normal physiological function is critical. Given the wide spread distribution it is often challenging to target specific subclasses of NMDA receptor. In the present manuscript the authors address a new or modified class of pharmacological agents targeting NMDA receptors, the EU1794 series. Surprisingly certain members of this series can be interconverted from negative allosteric modulators (NAMs) to positive allosteric modulators (PAMs), or vice versa, by small changes in their chemistry. The authors do an outstanding job of characterizing these compounds and their properties that would be important to their potential use as experimental tools. The authors show that the activity of the compounds is dependent on agonist concentrations. Notable here is that some of the agents have no or only small modulatory effects at saturating concentrations but have very robust effects at sub-saturating concentrations. The authors characterize the time course of action showing that the compounds require glutamate and glycine binding before the compounds can have their modulatory effect. They identify potential sites of action for both negative and positive modulators and propose that they share overlapping sites.

The manuscript is well laid out and written-well. The conclusions are consistent with the presented data.

My only major comment is why do subtle changes in the structure of the EU1794 compounds lead to variations in modulation, either negative or positive allosteric modulation? Although the authors do a nice job of characterizing the effect of the compounds, it would be nice to have some sense of why the variation. Although the new compounds and the interconversion is interesting, it is not clear either structurally or mechanistically why they are interconverted.

Reviewer #3:

The manuscript by Perszyk et al., describes the work that involves discovery of positive and negative allosteric modulators (PAM and NAM) on NMDA receptors and characterization of the functional effects elicited by the compound binding. The compounds presented in this paper are novel in terms of compound-design and functional effects. The importance of PAM and NAM compounds for NMDA receptors in the field of neuropharmacology has been well illustrated for many years owing to their therapeutic potentials. The effort started a while ago with ifenprodil that binds an amino terminal domain (NTD) of NMDA receptors in a subtype-specific manner. In recent years, more efforts in developing a series of such compounds along with a completely new series of PAMs and NAMs have been undertaken in the field, thus, the importance of compound development for NMDA receptors is undisputable. The authors synthesized and purified enantiomers of the EU compounds and conducted thorough experiments to prove (at least at the level of pharmacology) that the binding sites for the PAM and NAM overlap by Schild-like analysis and demonstrated that a subtle difference in chemical structures can convert PAM to NAM and vice versa. They also demonstrated the binding site to be located at the tip of the transmembrane domain by extensive mutagenesis and domain truncations experiments. The overall quality of the work is high and it represents novel findings in a variety of ways (discovery of novel compounds, characterization). This manuscript contains useful information from chemistry perspective as well. Overall the work well qualifies to be published in *eLife*.

Results in Figure 6, especially the point mutations that reversed the effect of NAM is robust and interesting. I wonder if the authors can say more about the potential mechanism for PAM-NAM conversion. Related to this, it would be good to have a rough estimation of what the mutations do to the ion channel activity for the ones that inversed the activity. For example, what does 1a F654 do in terms of P open (rough estimation) and agonist potency since they both affect PAM/NAM activity as authors implied.

---

## [Author Response]

Reviewer #2:[…] The manuscript is well laid out and written-well. The conclusions are consistent with the presented data.My only major comment is why do subtle changes in the structure of the EU1794 compounds lead to variations in modulation, either negative or positive allosteric modulation? Although the authors do a nice job of characterizing the effect of the compounds, it would be nice to have some sense of why the variation. Although the new compounds and the interconversion is interesting, it is not clear either structurally or mechanistically why they are interconverted.

We are extremely interested in this question and are working with Hiro Furukawa to obtain structural information that would speak to the structural determinants of positive and negative allosteric modulation. This project has a long time-line, beyond the current study, as we are just now initiating co-crystallization screens.

Reviewer #3:[.] Results in Figure 6, especially the point mutations that reversed the effect of NAM is robust and interesting. I wonder if the authors can say more about the potential mechanism for PAM-NAM conversion. Related to this, it would be good to have a rough estimation of what the mutations do to the ion channel activity for the ones that inversed the activity. For example, what does 1a F654 do in terms of P open (rough estimation) and agonist potency since they both affect PAM/NAM activity as authors implied.

We are very interested in the mechanism of interconversion, and only have speculative hypotheses at present. We believe we need structural information to address this idea and have initiated a collaboration with a crystallographer to obtain data describing the site of action. We also have a computational program to dock ligands into space near the pre-M1 region. Neither project has progressed to a point where there is publication-ready data. However, to begin to explore ideas around this interconversion of modulator direction, we have added new data to the manuscript describing the effect of various mutations on the open probability. These data show various alterations in open probability for homologous mutations that interconvert the effect of the modulator, however no direct correlation is observed between the alteration in open probability and the alteration of modulator actions. We also continue to synthesize new derivatives, which could yield additional clues about nature of this interconversion.